# A Review of the T-Stub Components for the Analysis of Bolted Moment Joints

**Giovani Jesus Berrospi Aquino** [1], **Ana María Gómez Amador** [2], **Jorge Hernan Alencastre Miranda** [1] and **Juan José Jiménez de Cisneros Fonfría** [1,*]

1    Facultad de Ciencias e Ingeniería, Pontificia Universidad Católica del Perú, Lima 15088, Peru; giovani.berrospia@pucp.pe (G.J.B.A.); jalenca@pucp.edu.pe (J.H.A.M.)
2    Department of Mechanical Engineering, Universidad Carlos III de Madrid, 28911 Leganés, Spain; amgomez@ing.uc3m.es
*    Correspondence: juanjose.cisneros@pucp.pe; Tel.: +51-997513334

**Abstract:** The analytical method for the calculation of the properties of a bolted joint established by the structural Eurocodes proposes the T-stub as a component for the characterization of the tension and compression zones in moment joints. In this article, a review of the state of the art on the T-stub component is developed, where the works developed since it was initially defined, and from the perspectives of formulation, experimentation and numerical simulation are summarized and discussed. Additionally, possible future lines of work are proposed.

**Keywords:** T-stub; components method; Eurocodes; numerical simulations; experimental tests

## 1. Introduction

Over the decades, steel structures have been the preferable option to construct building ranging from small houses to skyscrapers. This predilection is because of the overwhelming benefits of the material such as strength, uniformity, elasticity, ductility, toughness, etc. These benefits give steel structures the advantages of being lighter (in weight per length) than concrete or wood and making it easy to modify or make amplifications. Nevertheless, steel structures have the disadvantages of quickly corroding in the typical environment and susceptibility to fatigue or buckling, etc.

According to Eurocode 3 [1], steel structures can be classified into braced frames and unbraced frames. Braced frames have a system of stiff elements called bracings, which is provided to withstand the sum of the lateral forces. In the other case, the unbraced frame is adopted [2]. Additionally, the standard provided a further classification based on the sensitivity to second order effects in the elastic range. The term non-sway frame is used when the lateral-stiffness is sufficient to allow the geometrical second order effects to be neglected. On the other hand, when the lateral stiffness is not enough to neglect the geometrical second order effects, whether for braced or unbraced frames, the term sway frames is used [2]. Another type of classification divides them into steel trusses and steel portal frames, where the principal differences between them are the arrangement of elements and the type of connection. Usually, the trusses have triangular units of straight elements connected by pinned joints; they do not have rotational stiffness. On the other hand, the portal frames are made of shape profiles connected by joints traditionally considered rigid.

The most common portal frames moment connections are the T-stub connection (see Figure 1a), and the end-plate bolted connection (see Figure 1b). The T-stub connection is a type of connection where T profiles are connected from its web to the beam flange and from its flange to the column flange. The T-stub connection was first studied by Douty and McGuire [3].

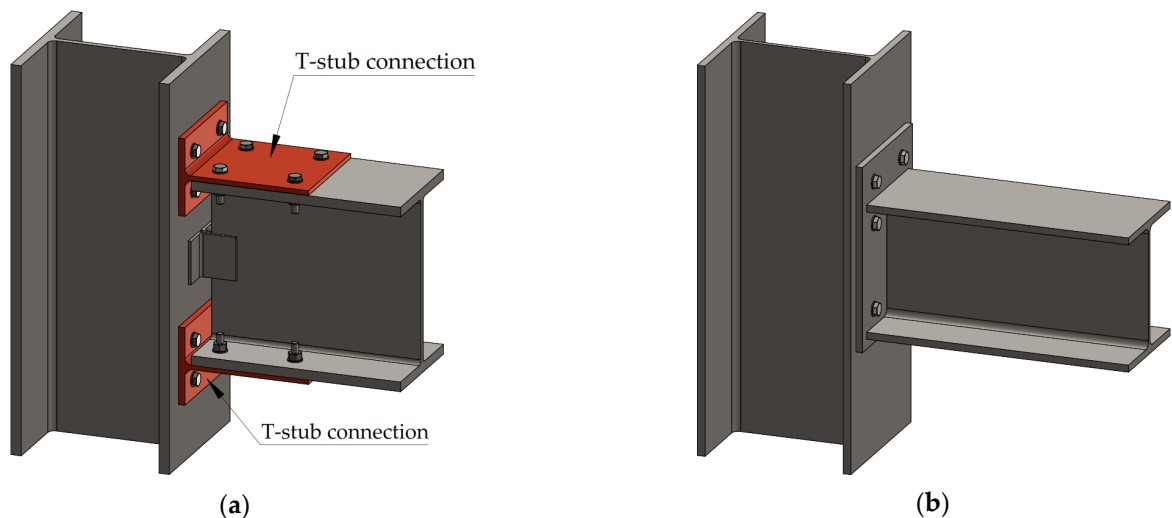

**Figure 1.** (**a**) T-stub connection; (**b**) End plate connection.

On the other hand, the end-plate bolted connection is a type of connection where a plate is welded to the cross-section of the beam, and then it is joint to the column flange employing bolts. Moreover, the end-plate bolted connection has more structural advantages because of the greater lever arm and because it permits placing a greater number of bolts near to the tension zone, which reduces the bending moment in the end-plate [3]. This type of joint is studied by the component method, Eurocode 3-1.8 [1], which considers the joint as a set of single basic components located in three different single zones: the tension zone, the compression zone and the shear zone, see Figure 2. These components of the joint can be represented in a mechanical spring model (see Figure 2b), where the characteristic of each spring are obtain from the properties of each component (Colum Web in Tension CWT, Colum Web in Compression CWC and Column Web in Shear CWS). The T-stub is the component that characterises the behaviour of the tension zone. The T-stub has an elastic behaviour at the beginning, see Figure 3b, because its elements do not reach the yield stress of the constitutive law, see Figure 3a, and the force-displacement curve of this zone is linear. However, when the stresses are superior to the yield stress, the T-stub elements begin to plasticize and the knee zone is observed before the slope of the curve changes, see Figure 3b.

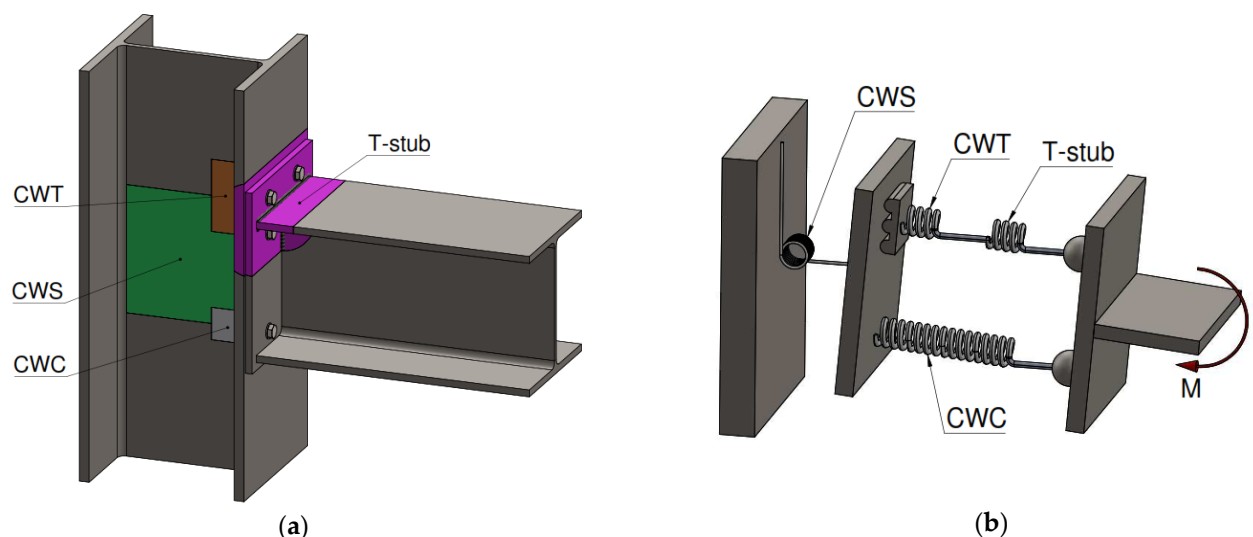

**Figure 2.** (**a**) Components; (**b**) Spring model.

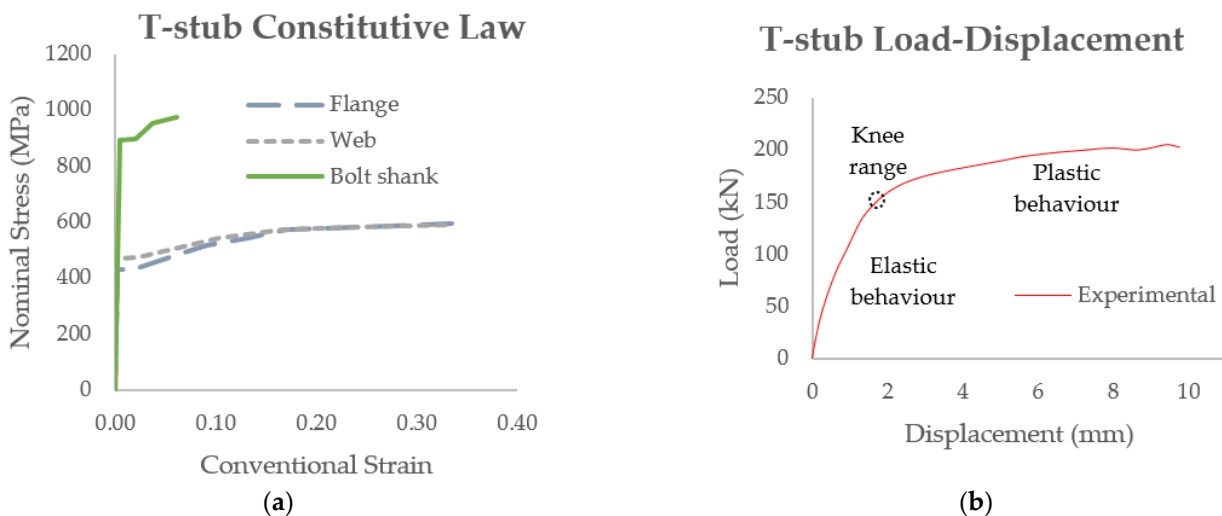

**Figure 3.** (**a**) T-stub constitutive law; (**b**) T-stub Load-Displacement [4].

Different authors have been studying the T-stub component since 1965 in three different manners: analytical, experimental and numerical simulations. As a result of these investigations, researchers got to the conclusion that the T-stub component can fail in three modes. Mode 1 happens when the flange reaches the yield stress, and the fracture of the flange follows without bolt failure. In Mode 2, the flange yields, and the bolts reach their ultimate stress. Mode 3 is produced when the flange works in the elastic zone and the bolts fail. The most important geometrical parameters are presented in Figure 4. For instance, "*r*" and "*a*" are the fillet radius and the throat of the weld, respectively.

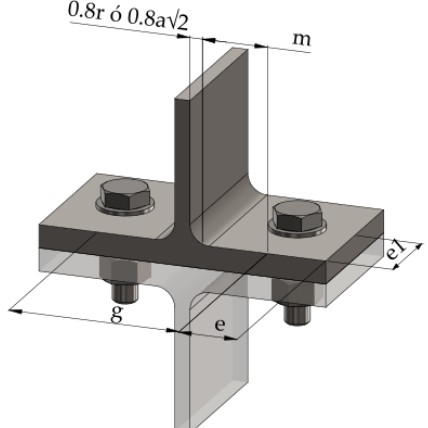

**Figure 4.** Geometrical parameters of T-stub component.

This article provides a review of state of the art of T-stub components since they were initially defined. Therefore, it is convenient to divide the state of the art into three categories: analytical, experimental and numerical simulation. These categories are organised in chronological order. This article also aims to propose future lines of work.

## 2. Analytical Model

The analytical models that are boarded in this part are developed by the two most important steel construction institutions: CEN and the AISC. The first one is in charge of developing the Eurocodes (European Union standards) such as Eurocode 3, which parts 1–8 are about the design of joints. A U.S. institution develops the last one, and it is in charge of the structural codes for the USA., such as the AISC 360, the manuals and the design guides. Additionally, other analytical models based on EC 3 are presented in this section.

### 2.1. Component Approach (EC3) T-Stub in Tension

The characterisation of the behaviour of a joint can be studied by the analytical approach, which is called the component method. The component method is a practical method initially used in EC 1992 that enables predicting the response of any steel or composite joint based on the knowledge of the mechanical and geometrical properties of the material [5]. As mentioned before, in the component method, a joint is considered a set of individual basic components. For this review, the component T-stub of the end-plate connection is considered, located in the tension zone, see Figure 2. The T-stub resistance is predicted by the analytical model found in the Eurocode 3 part 1–8 [1]. This prediction is made in nominal values, without the uses of safety factors. According to this standard, the resistance of the T-stub could be determined by three different failures modes:

(1) Mode 1: the flange of the T-stub reaches the yielding stress before the bolts get into the plastic zone. This type of behaviour could be classified as a pinned joint, see Equation (1) or Equation (2) and Figure 5a.
(2) Mode 2: this mode is characterised by the yielding of the flange of the T-stub and the failure of the bolts (bolt fracture). This type of behaviour could be classified as a semi rigid joint, see Equation (3) and Figure 5b.
(3) Mode 3: this mode is characterised by the failure of the bolt without the flange reach the yielding point. This type of behaviour usually happens when flanges are quite thick, and this type of behaviour could be classified as a rigid joint, see Equation (5) and Figure 5c:

$$F_{Rd,1} = \frac{4 \times M_{pl,1,\,Rd}}{m}, \tag{1}$$

$$F_{Rd,1} = \frac{(8n + 2e_w)M_{pl,1,\,Rd}}{2mn - e_w(m+n)}, \tag{2}$$

$$F_{Rd,2} = \frac{2M_{pl_2,\,Rd} + n\sum B_{t,Rd}}{m+n}, \tag{3}$$

$$M_{pl,i,\,Rd} = \frac{1/4\sum beff_i \times tf^2 \times f_y}{\gamma_{M0}}, \tag{4}$$

$$F_{Rd,3} = \sum B_{t,Rd}, \tag{5}$$

$$B_{t,Rd} = k_2 f_{ub} A_s / \gamma_{M2}, \tag{6}$$

where $F_{Rd,i}$ is the T-stub design resistance for each failure mode, $M_{pl,i,\,Rd}$ is the plastic flexural resistance of the T-stub based on the $beff$ (see Equation (7) and Figure 6) and $tf$, which are the effective width and the flange thickness. $e_w$, $m$ and $n$ are a quarter of the diameter washer, the bolt distance to the plastic hinge form near to the web and $n$ is the minimum distance to the edge, respectively, $n = e$ (see Figure 4) but $n \leq 1.25m$. $B_{t,Rd}$ is the bolt tension resistance, $k_2$ is the factor that takes a value of 0.68 for countersunk bolts and 0.9 for other cases, $A_s$ is the tensile area of the bolt. $f_y$ and $f_u$ are the yield and the ultimate stresses. Finally, $\gamma_{M0}$ and $\gamma_{M2}$ are the partial factors for design and take the values of 1 and 1.25, respectively.

$$beff = \min(beff_1, beff_2, beff_3, beff_4, beff_5) = \begin{cases} beff_1 = 2\pi m \\ beff_2 = 4m + 1.25e \\ beff_3 = \pi m + 0.5p \\ beff_4 = 2m + 0.625e + 0.5p \\ beff_5 = b \end{cases}, \tag{7}$$

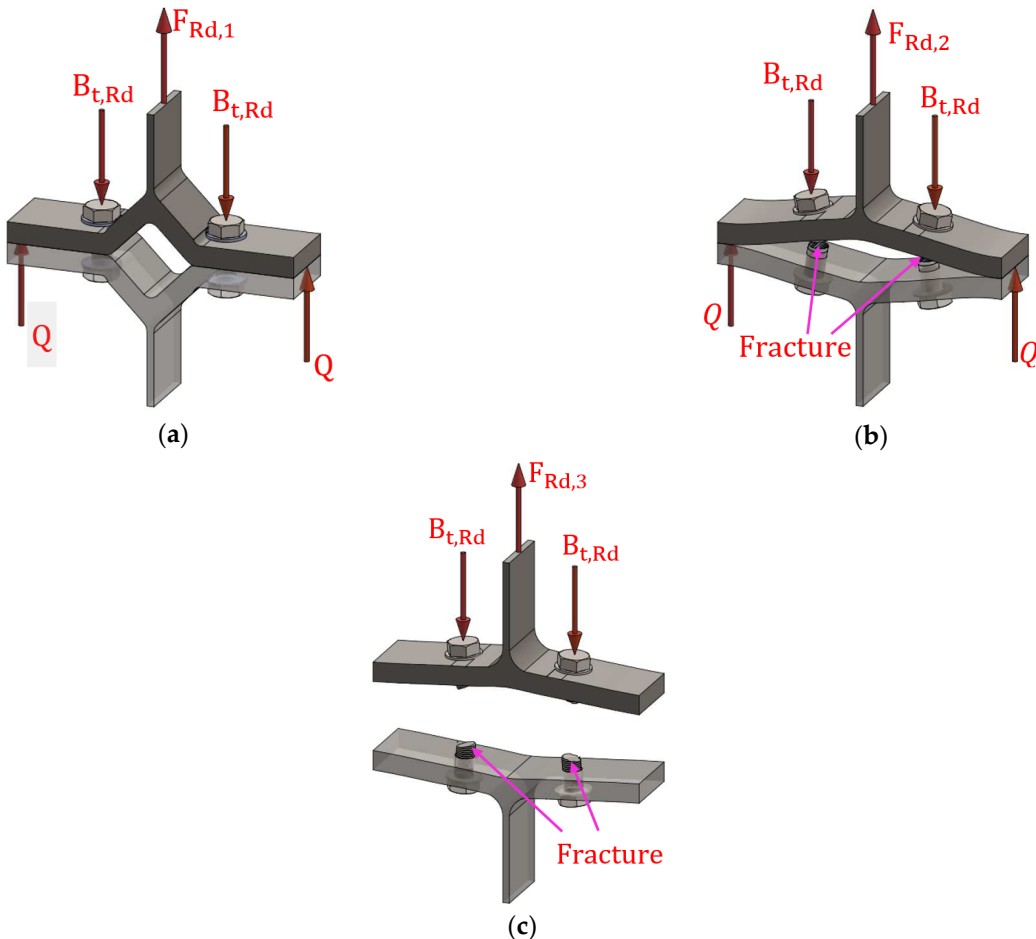

**Figure 5.** T-stub Failure modes: (**a**) Mode 1; (**b**) Mode 2; (**c**) Mode 3.

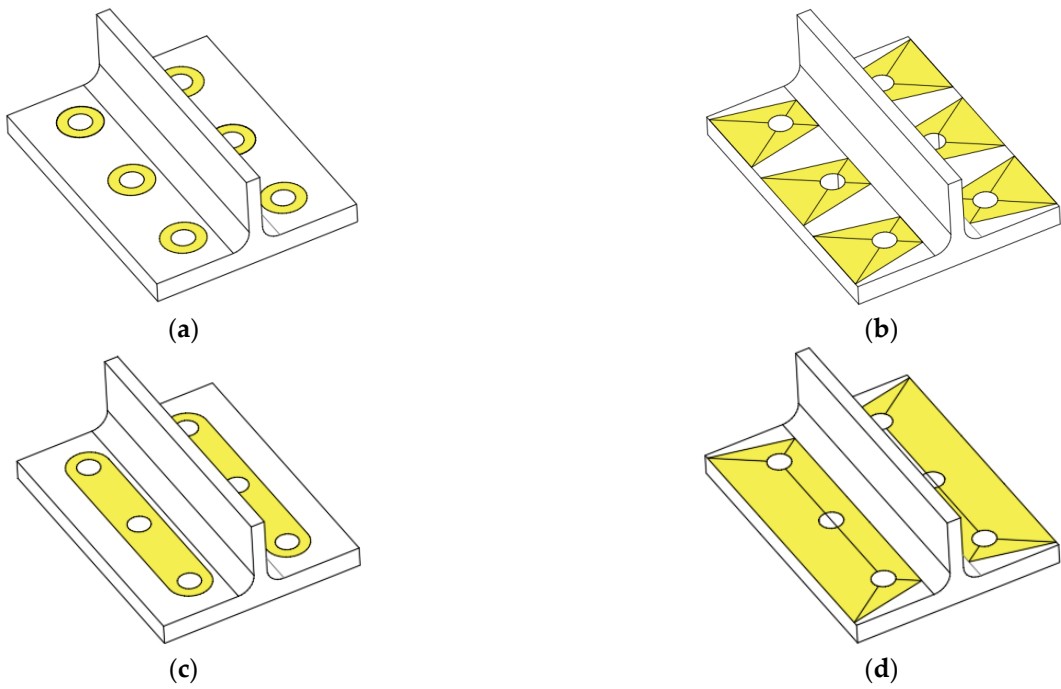

**Figure 6.** *Cont.*

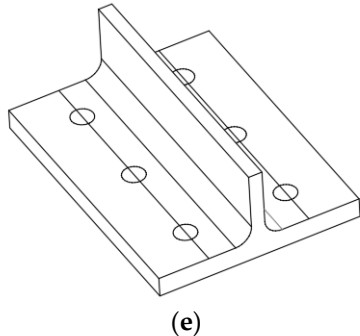

**(e)**

**Figure 6.** Patterns of effective width: (**a**) Three bolt row considered individually-circular pattern $beff_1$; (**b**) Three bolt row considered individually-non circular pattern $b_{eff_2}$; (**c**) Group of bolt row- circular pattern $beff_3$; (**d**) Group of bolt row- non circular pattern $beff_4$; (**e**) Beam pattern $beff_5$.

The initial stiffness of the component was determined by Jaspart [6] by considered the T-stub as a simple supporting beam, the supports corresponding to the location of the prying forces, see Equations (8)–(11) and Figure 7. Furthermore, Jaspart [7] and Jaspart and Maquoi [8] determined the plastic stiffness by assuming a linear relationship, see Equations (11) and (12):

$$k_{e,T,u,l} = \frac{E(0.9beff)tf_{u,l}^3}{m_{u,l}^3},$$ (8)

$$k_{e,bt} = 1.6\frac{EA_s}{L_b},$$ (9)

$$L_b = t_{fu} + t_{fl} + 2t_{wsh} + 0.5(t_n + l_n),$$ (10)

$$k_{e,o} = \frac{1}{\frac{1}{k_{e,T,u}} + \frac{1}{k_{e,T,l}} + \frac{1}{k_{e,bt}}},$$ (11)

$$k_{pl} = \frac{2(1+v)}{3}\frac{E_h}{E}k_{e,o},$$ (12)

where $E$ is the Young's modulus, $u, l$ refers to the upper and low T-stub, $A_s$ refers to the effective resistance area of the bolt, $L_b$ is the conventional bolt length, $k_{e,o}$ is the initial stiffness, $v$ is the Poisson's modulus, $E_h$ is the strain hardening modulus.

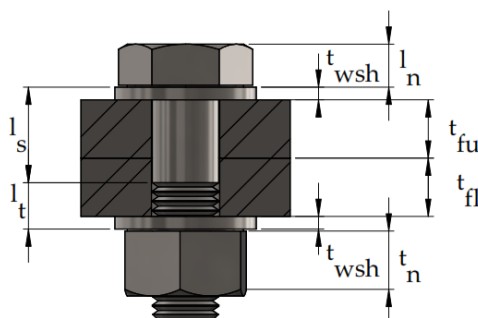

**Figure 7.** Bolt geometrical properties.

*2.2. Component Approach (EC3) T-Stub in Tension with Backing Plates*

The resistance of T-stub with backing plates, see Figure 8, is given in the EC for the failure mode 1, see Equations (13) and (14). However, the European standard does not cover the increase of the stiffness of T-stub due to the use of backing plates.

$$F_{Rd,1} = \frac{4 \times M_{pl,1, Rd} + 2M_{bp, Rd}}{m},$$ (13)

$$F_{Rd,1} = \frac{(8n - 2e_w) \times M_{pl,1,\,Rd} + 4nM_{bp,\,Rd}}{2mn - e_w(m + n)}, \tag{14}$$

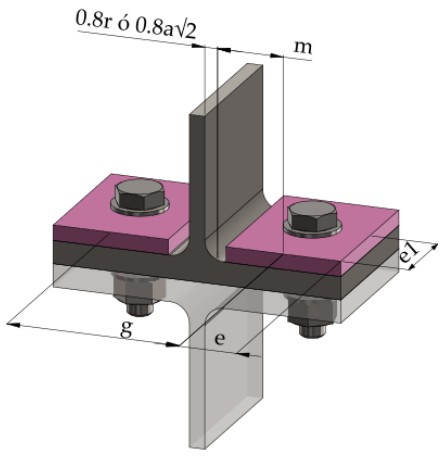

**Figure 8.** Geometric properties of T-stub component with backing plate.

### 2.3. Analytical Model for T-Stub with Four Bolt Per-Row Based on EC3

T-stub with four bolts per-horizontal row, see Figure 9, analytical model was studied by Demonceau et al. [9,10]. They proposed a variation of the formula for the failure Mode 1, where the parameter of $n = e$ change to $n = e_1 + e_2$, and for failure Mode 2, see Equations (15)–(17), where $n_1 = e_1$ and $n_2 = e_2$.

$$F_{Rd,2,p} = \frac{2M_{pl_2,\,Rd} + \frac{\sum B_{t,Rd}}{2} \times \left(n_1^2 + 2n_2^2 + 2n_1 n_2\right)}{m + n_1 + n_2}, \tag{15}$$

$$F_{Rd,2,np} = \frac{2M_{pl_2,\,Rd} + \frac{\sum B_{t,Rd}}{2} \times (n_1)}{m + n_1}, \tag{16}$$

$$F_{Rd,2} = \min\left(F_{Rd,2,p}, F_{Rd,2,np}\right), \tag{17}$$

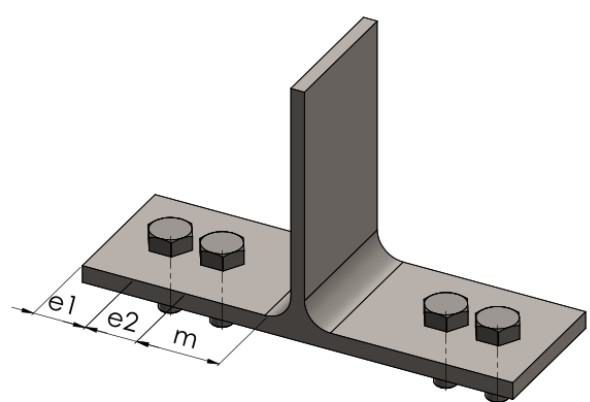

**Figure 9.** T-stub geometry configuration-four bolts per row.

### 2.4. AISC Analitycal Model

The AISC manual [11] provides formulas for determining the required thickness of the T-stub flange, see Figure 10. The minimum flange thickness $t_{min}$ to avoid the prying

force effects is given in Equation (18). Nevertheless, the required thickness to consider the prying force effects is produced when $t \leq t_{min}$, see Equation (19):

$$t_{min} = \sqrt{\frac{4Bb'}{pF_u}},$$

(18)

$$t_{required} = \sqrt{\frac{4Tb'}{pF_u(1+\delta)}},$$

(19)

where in Figure 10 and Equations (18) and (19), $B = 0.75F_uA$ is the bolt tension capacity and T is the half of the applied axial load and p is de tributary length, $F_u$ is the flange ultimate stress, $b' = b - d_b/2$; $d_b$ is the bolt diameter, $\delta = 1 - d/p$, see [11].

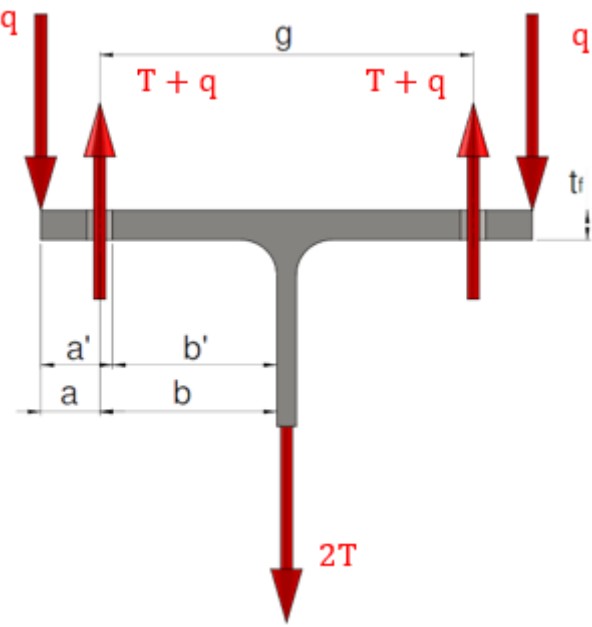

**Figure 10.** AISC T-stub model.

### 2.5. Proposed Analytical Model for T-Stub under Large Deformation

The proposed analytical model for T-stub under large deformation is based on EC3 and was developed by Tartaglia et al. [12] to estimate the ultimate resistance of T-stub connections under large deformation due to gap openings. Thus, it is suitable for connections with $\beta$ (see Equation (20)) parameter not greater than 1.2. Besides, for connections designed for Mode 1 and 2. Moreover, the proposed model is characterized for combine the primary effects due to gap opening in small deformations (see EC3) with the secondary effects due to the development of the membrane action and its corresponding pattern of plastic deformations. It is important to mention that the analytical model took into account three secondary resistance mechanisms: the tensile yielding of the flange with its resistance R1, the bearing of the bolt to edge with its resistance R2 and the failure of the bolts under shear force, axial force and bending moment with its resistance R3. The equations that estimate the resistance under large deformation are given as follows:

$$\beta = \frac{4M_{pl,Rd}}{(m\sum F_{t,Rd})},$$

(20)

$$F_T = F_{T,1} + F_{T,LD} \, for \, mode1,$$

(21)

$$F_T = F_{T,2} + F_{T,LD} \, for \, mode2,$$

(22)

where $F_{T,1}$ and $F_{T,2}$ are the resistance for failure Mode 1 and 2 (according to EC3), while $F_{T,LD}$ is the additional contribution due to the development of large deformations, and it is determined using Equation (23):

$$F_{T,LD} = 2 \times \min(R_1, R_2, R_3) = 2 \times \min(\frac{b_{eff}}{2} t_f f_u \sin(\alpha); F_{B,R} \tan(\alpha); F_{t,R} \times \psi), \quad (23)$$

where $b_{eff}$, $t_f$, $f_u$ and $F_{B,R}$ are the effective length, the flange thickness, the steel ultimate strength of the flange and the bolt to edge bearing resistance, respectively. The angle $\alpha$ is the ratio between the imposed gap opening and the distance $m$; $\psi$ is the reduction factor to decrease the tensile resistance of the bolts to account for the interaction with shear force and bending moment, ranging within 0.3–0.4 for mode 1 and 0.4–0.6 for Mode 2.

### 2.6. Component Approach (EC9) Aluminium T-Stub in Tension

The aforementioned component approach of the EC3 formulae are only applied for steel mild materials and not for other types of materials. Therefore, the standard EC9 [13] covered the design of aluminum structures and there is found the equations that predict the failures modes of aluminium T-stub components. These failure modes are quite similar to those proposed by EC3, but with a difference in Mode 2, which is divided into two failure modes:

(1) Mode 1: the flange failure by developing 4 plastic hinges, 2 of them at the web-flange intersection ($w$) and the other 2 at the bolt location ($b$), see Equation (24).
(2) Mode 2a: flange failure by developing 2 plastic hinges with bolt forces at the elastic limit, see Equation (25).
(3) Mode 2b: bolt failure with yielding of the flange at the elastic limit, see Equation (26).
(4) Mode 3: bolt failure and the flange is in the elastic zone, see Equation (27).

$$F_{Rd,1} = \frac{2(M_{u,1})_w + 2(M_{u,1})_b}{m}, \quad (24)$$

$$F_{Rd,2a} = \frac{2M_{u,2} + n \sum B_o}{m + n}, \quad (25)$$

$$F_{Rd,2b} = \frac{2M_{o,2} + n \sum B_u}{m + n}, \quad (26)$$

$$F_{Rd,3} = \sum B_u, \quad (27)$$

$$M_{u,1} = 0.25 t_f^2 \times beff_1 \times \rho_{u,haz} \times f_u \times \frac{1}{k} \times \frac{1}{\gamma_{M1}}, \quad (28)$$

$$M_{u,2} = 0.25 t_f^2 \times beff_2 \times \rho_{u,haz} \times f_u \times \frac{1}{k} \times \frac{1}{\gamma_{M1}}, \quad (29)$$

$$M_{o,2} = 0.25 t_f^2 \times beff_2 \times \rho_{o,haz} \times f_o \times \frac{1}{\gamma_{M1}}, \quad (30)$$

$$\frac{1}{k} = \frac{f_o}{f_u}(1 + \Psi \frac{(f_u - f_o)}{f_o}), \quad (31)$$

$$\Psi = \frac{(\varepsilon_u - 1.5\varepsilon_o)}{1.5(\varepsilon_u - \varepsilon_o)}, \quad (32)$$

$$\varepsilon_o = \frac{f_o}{E}, \quad (33)$$

$$B_o = \begin{cases} 0.9 f_y \times A_s / \gamma_{M2}, & \text{for steel bolts} \\ 0.6 f_o \times A_s / \gamma_{M2}, & \text{for aluminium bolts} \end{cases}, \quad (34)$$

In Equation (24), $(M_{u,1})_w$ should be determined by considering $\rho_{u,haz} < 1$ and $(M_{u,1})_b$ should be determined by considering $\rho_{u,haz} = 1$. From Equation (28) to Equation (30)

$\rho_{u,haz}$ and $\rho_{o,haz}$ are equal to 1 if not welded in a section. Furthermore, $f_o$, $f_u$, $f_y$ are the characteristic values of 0.2% proof strength, the characteristic value of ultimate tensile strength and the characteristic value of yield strength, respectively. $B_o$ and $B_u$ are the conventional bolt strength at elastic limit and the tension resistance of a bolt plate assembly. $\varepsilon_u$ and $A_s$ are the ultimate strain of the flange material, and the stress area of a bolt. Finally, $\gamma_{M1}$, $\gamma_{M2}$ are the partial factors for design and take the values of 1 and 1.25, respectively.

### 2.7. Proposed Analytical Model for Clamped T-Stub Based on EC3

The proposed analytical model for a clamped T-stub (see Figure 11) is based on the EC3 and was developed by Cabaleiro [14] in 2016. This type of T-stub, as in EC3, has three failure modes, which are calculated with Equations (35)–(38):

$$F_{T,m1} = \frac{beff \times t^2 \times f_y}{m \times \gamma_{M0}}, \tag{35}$$

$$F_{T,m2} = \frac{n \times \frac{F_{t,Rd \times b}}{a+b} + beff \times t^2 \times \frac{f_y/\gamma_{M0}}{4}}{(m+n)/2}, \tag{36}$$

$$F_{T,m3} = \frac{2B_{t,Rd} \times b}{a+b}, \tag{37}$$

$$beff = 2m + h, \tag{38}$$

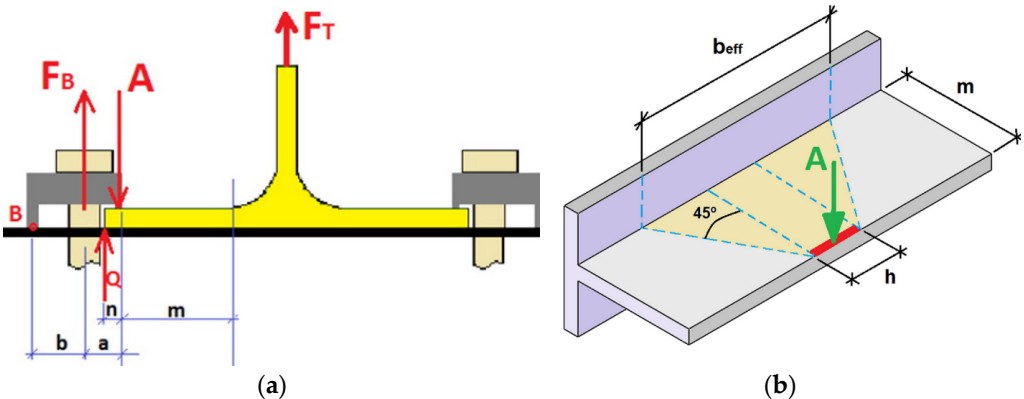

**Figure 11.** Clamped T-stub: (**a**) geometry and force action: (**b**) Action of the clamp on the flange [14]. "Reproduced with permission from Manuel Cabaleiro, Belén Riveiro, Borja Conde, José C. Caamaño, Journal of Constructional Steel Research; published by Elsevier, 2016".

### 2.8. Proposed Analytical Model for Blind Bolted T-Stub Based on EC3

This proposed model is based on EC 3 equations. In this analytical model Wang [15] proposed an analytical manner of determining the initial stiffness of the blind bolt, which is influence by the presence of the flaring sleeves. Therefore, the analytical model has two springs instead of one, see Figure 12a,b and Equations (39) and (40).

$$k_{bsl} = \frac{t_s A_{slp}}{\left[ v S_1^2 C_3 - S_2^2 \left( C_1 - \frac{v}{2} C_2 \right) \right] \sin(\alpha)}, \tag{39}$$

$$k_b = \frac{1}{k_{bsh} + k_{bsl}}, \tag{40}$$

where $k_{bsl}$ is the stiffness of the flaring sleeve, $k_b$ is the total stiffness of the hollo-bolt $t_s$ is the thickness of the sleeve, $v$ is the Poisson modulus, $C_1 = \cos^2(\alpha) \cot(\alpha)$, $C_2 = \cot(\alpha)$, $C_3 = \frac{1}{v} C_1 - \frac{1}{2} C_2(\alpha)$, $k_{bsh}$ is equal to the stiffness of the bolt shank of EC 3, see Equation (9). Additionally, the effective contact area is calculated by $A_{spl} = \gamma \pi \left( d_{tcm}^2 - d_{tct}^2 / 4 \right)$ and $\alpha$, $d_{tcm}$, $d_{tct}$ are shown in Figure 13.

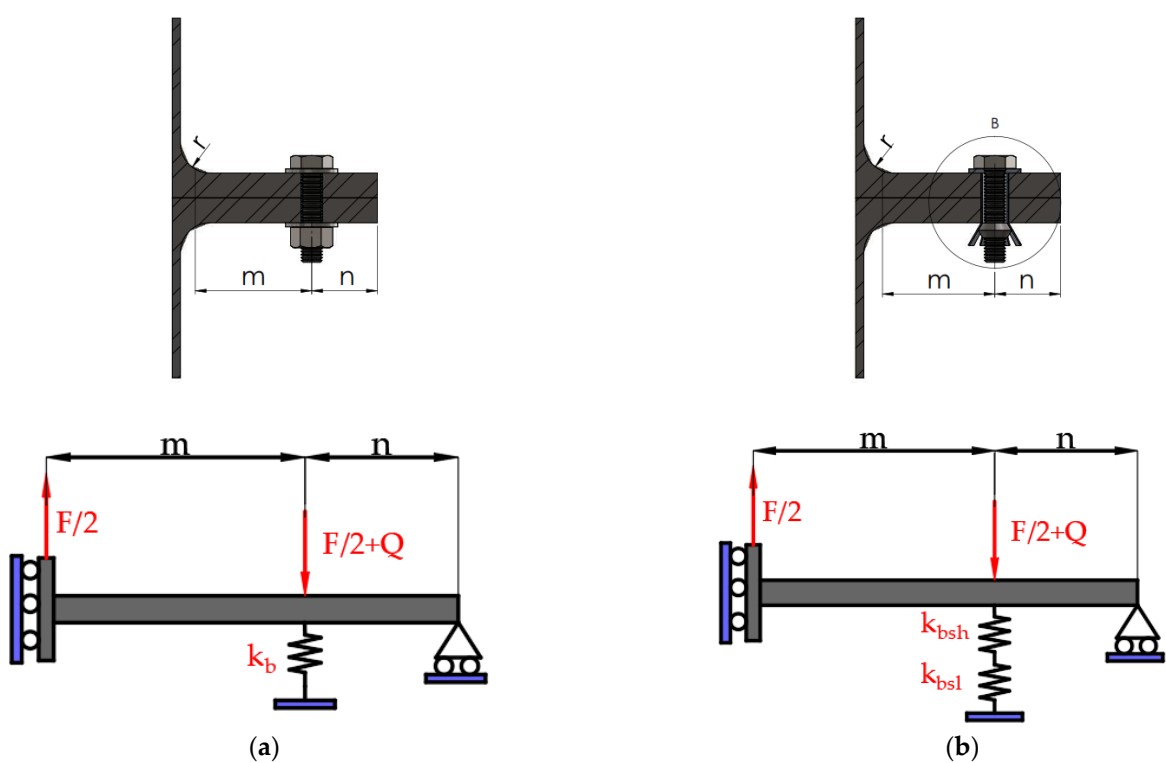

**Figure 12.** Analytical model of T-stub connection: (**a**) Standard model; (**b**) Modified model for hollo-bolt [15]. "Reproduced with permission from Z.Y. Wang, W. Tizani, Q.Y. Wang, Engineering Structures; published by Elsevier, 2010".

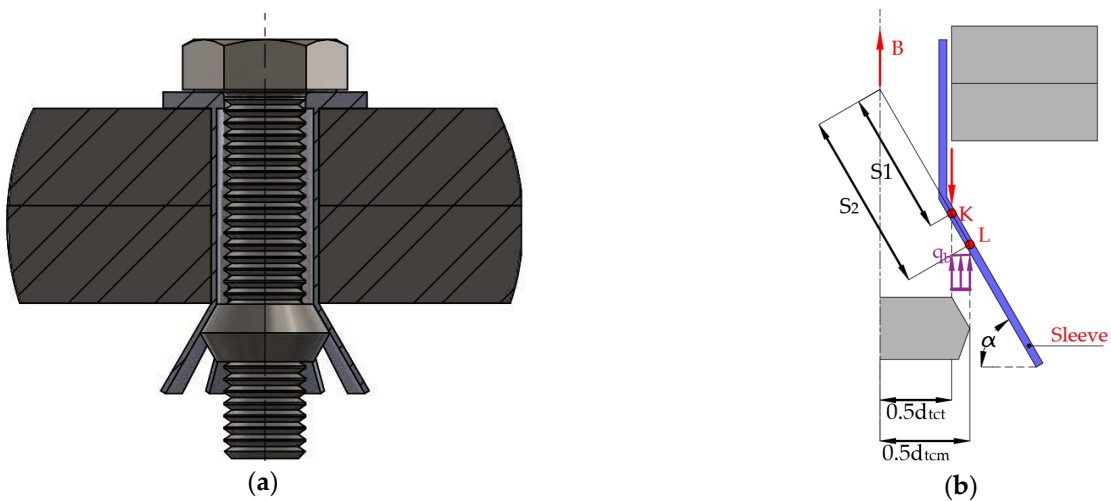

**Figure 13.** Assemble of hollo-bolt connection: (**a**) Hollo-bolt assemble; (**b**) Schematic representation of mechanical interlocking load acting on the flaring sleeve [15]. "Reproduced with permission from Z.Y. Wang, W. Tizani, Q.Y. Wang, Engineering Structures; published by Elsevier, 2010".

### 2.9. Proposed Analytical Model for One-Side Bolted T-Stub Based on EC3

The one-side bolted T-stub has five failure modes, where the Mode 1, Mode 2a, Mode 3a were analysed by the EC 3, and the other two Mode 2b and Mode 3b were proposed by et al. [16], Wulan et al. [17], Zhu et al. [18], see Figure 14.

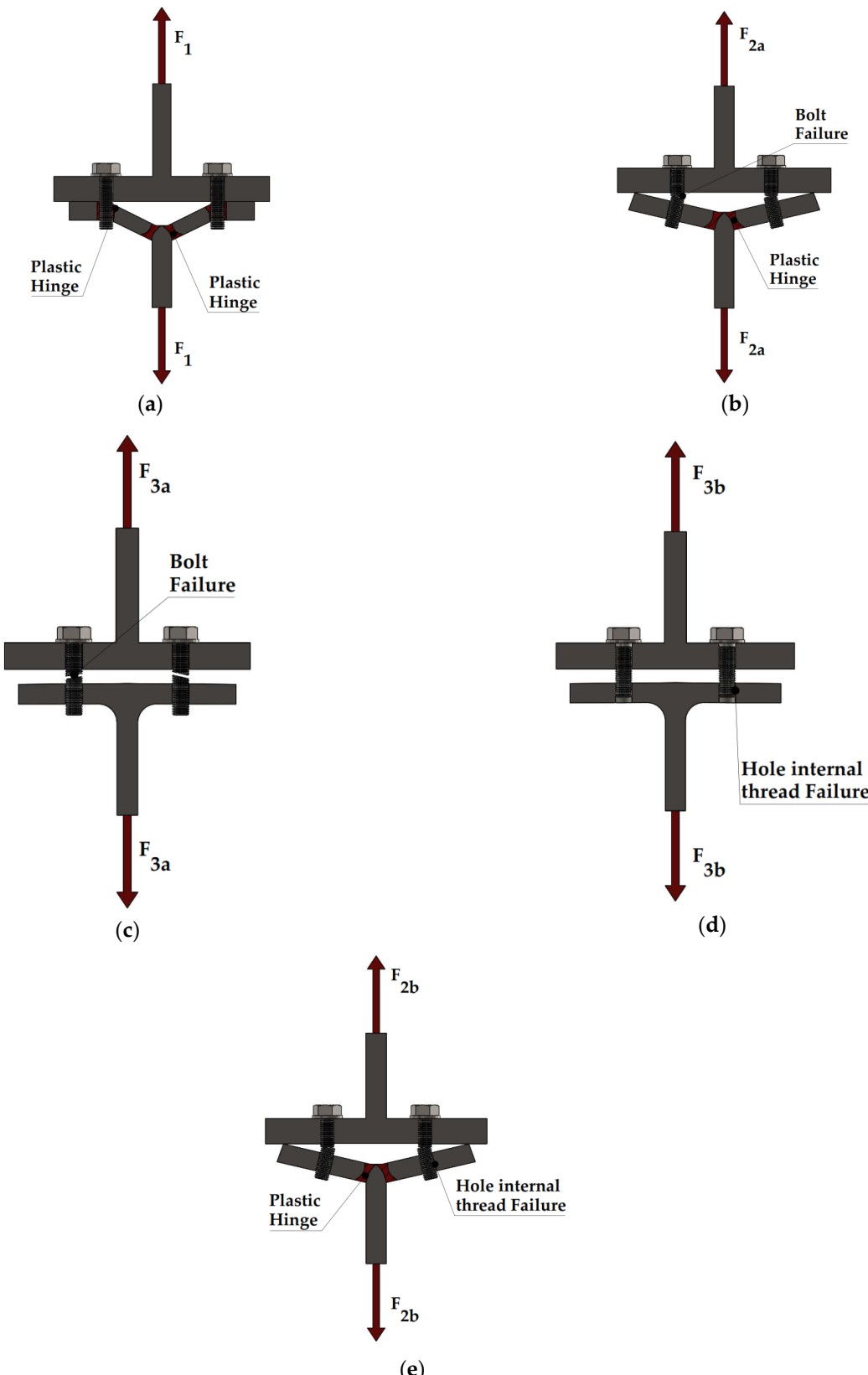

**Figure 14.** Failure modes of one-side bolted T-stub: (**a**) Mode 1; (**b**) Mode 2a; (**c**) Mode 3a; (**d**) Mode 3b; (**e**) Mode 2b [17]. "Reproduced with permission from Tuoya Wulan, Peijun Wang, Yang Li, Yang You, Funing Tang, Engineering Structures; published by Elsevier, 2018.

(1)  Mode 1: This type of failure mode is the same than the EC 3, and is calculated by Equation (1), but the plastic moment, see Equation (4), is determined by Equation (41), where $f_t$ is the ultimate strength of the T-stub flange.

(2)  Mode 2a: This failure is the same as the EC 3 Mode 2, see Equation (3), but plastic moment is calculated as for Mode 1, see Equation (41).

(3)  Mode 3a: The bolt failure characterises this mode, and it is the same as EC3, see Equations (5) and (6)

(4)  Mode 3b: The hole thread failure characterises this failure mode. This tension strength could be calculated by Equations (42)–(48).

(5)  Mode 2b: The flange yielding accompanied with hole thread failure characterize this failure mode, and the tension strength could be calculated by Equations (49) and (50):

$$M_{pl,i, \; Rd} = \frac{1/4 \sum beff_i \times tf^2 \times f_t}{\gamma_{M0}}, \tag{41}$$

$$F_{T,3b} = \sum F_{s,Rd}, \tag{42}$$

$$F_{s,Rd} = \min\left(\sum F_{s,1,Rd}, \sum F_{s,2,Rd}\right), \tag{43}$$

where $F_{s,1,Rd}$ is the shear strength of one thread and $F_{s,2,Rd}$ is the bending strength of one thread.

$$F_{s,1,Rd} = A_v f_{yv,p}, \tag{44}$$

$$A_v = \pi D h_s, \tag{45}$$

$$F_{s,2,Rd} = \frac{W_p f_{y,p}}{b_s}, \tag{46}$$

$$W_p = 0.25 \pi D h_s^2, \tag{47}$$

$$f_{yv,p} = f_y / \sqrt{3}, \tag{48}$$

$f_{yv,p}$, $f_{y,p}$ and $f_y$ are the shear strength and yield strength of the T-stub flange and the yield strength of the steel. $A_v$ is the efficient shear area of one circle of threads. $D$, $b_s$ and $h_s$ are the external diameter of the bolt thread, the height and width of the internal thread on flange, respectively, which depends on the type of the thread:

$$F_{T,2b} = \frac{2M_{pl,2b, \; Rd} + n \sum F_{s,Rd}}{m + n}, \tag{49}$$

$$M_{pl,2b, \; Rd} = \frac{1/4 \sum beff_{2b} \times tf^2 \times f_y}{\gamma_{M0}}, \tag{50}$$

*2.10. Proposed Analytical Model for T-Stub under Impact Loading Based on EC3*

In 2015, Ribeiro et al. [19] proposed an analytical model based on EC3 and the yield line analytical model developed by Yu et al. [20] to predict the behaviour of T-stubs under impact loads due to blasts or impacts. The proposed method considered the enhancement of the constitutive law of the material by employing a dynamic increase factor (DIF), which will promote the increase of the elastic and ultimate strengths based on the ratio of the strength observed dynamically ($\sigma_{dyn}$) and statically ($\sigma$):

$$DIF = \frac{\sigma_{dyn}}{\sigma}, \tag{51}$$

then according to the Johnson-Cook model [21], Equation (52) is used to describe the $DIF_{steel}$ for intermediate strain rate values:

$$\sigma = [A + B\varepsilon^n] \times \left[1 + C \times ln\dot{\varepsilon}^*\right] \times [1 - (T^*)^m], \tag{52}$$

where $A$ is the quasi-static yield strength; $B$ and $n$ represent the effects of strain hardening; $m$ is the thermal softening fraction; $T^*$ is a non-dimensional parameter depending on the melting and transition temperatures to take into account the material softening due to a temperature variation (this is not considered in the proposed model); $\varepsilon$ is the equivalent plastic strain; $\dot{\varepsilon}^* = \dot{\varepsilon}/\dot{\varepsilon}_o$ is the dimensionless plastic strain rate, where $\dot{\varepsilon}$ is the strain rate and $\dot{\varepsilon}_o$ the reference quasi-static strain rate ($\dot{\varepsilon}_o = 0.001 s^{-1}$), $C$ is the strain rate constant. Then DIF could be calculated with Equation (53):

$$DIF = \left[1 + C \times ln\dot{\varepsilon}^*\right], \tag{53}$$

Additionally, the DIF value of the steel (for S355-C = 0.0039) and bolt (Bolt(8.8)-C = 0.0072) could be determined with the aid of Figure 15, where the values of $C$ are 0.0039 and 0.0072 for steel and bolt, respectively.

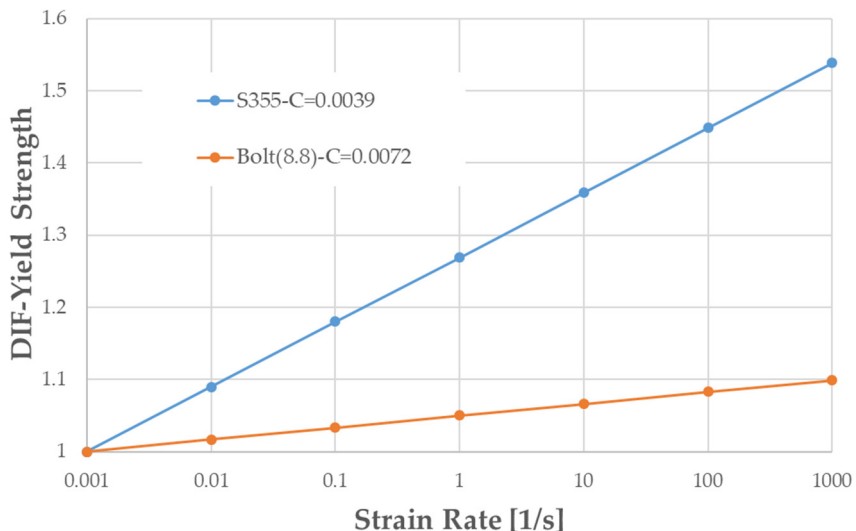

**Figure 15.** Dynamic increase factor (DIF) of the yield strength as function of the strain-rate [19]. "Reproduced with permission from João Ribeiro, Aldina Santiago, Constança Rigueiro, Luís Simões da Silva, Journal of Constructional Steel Research; published by Elsevier, 2015".

The failure modes that govern the T-stub behaviour are calculated with the same equations of EC3 1–8, see Equations (1), (3) and (5), but $f_y$ and $f_{ub}$ change to $f_{y,D}$ and $f_{ub,D}$, respectively, see Equations (54) and (55). $DIF_{steel}$ and $DIF_{bolt}$ could be considered equal to 1.5 and 1.1, for strain rates over $600 \ s^{-1}$, respectively:

$$f_{y,D} = DIF_{steel} * f_y, \tag{54}$$

$$f_{ub,D} = DIF_{bolt} * f_{ub}, \tag{55}$$

### 2.11. Proposed Analytical Model for Asymmetric T-Stub Based on Stiffness Matix

This analytical model, see Figure 16, was proposed by Jiménez de Cisneros [22] in 2016, and it is based on the stiffness matrix, see Equation (56), and is developed for the study of asymmetric T-stubs, which does not have a symmetric plane in the middle of the web due to the bolts position from this mid plane is not the same. The author considered that the conventional bolt length $L_b$ is calculated according to the Equation (10), but the

divided by two because just the upper half of the T-stub is analysis due to the symmetric conditions.

$$
\begin{bmatrix} 0 \\ 0 \\ 0 \\ F_t \\ 0 \\ 0 \\ 0 \\ 0 \\ 0 \end{bmatrix} = \frac{EI_f}{1+\Phi} \begin{bmatrix} k_{21} & k_{22} & k_{23} & 0 & 0 & 0 & 0 & 0 \\ k_{31} & k_{32} & k_{33} & k_{34} & k_{35} & 0 & 0 & 0 \\ 0 & k_{42} & k_{43} & k_{44} & k_{45} & k_{46} & 0 & 0 \\ 0 & k_{52} & k_{53} & k_{54} & k_{55} & k_{56} & k_{57} & 0 \\ 0 & k_{62} & k_{63} & k_{64} & k_{65} & k_{66} & k_{67} & 0 \\ 0 & 0 & 0 & k_{74} & k_{75} & k_{76} & k_{77} & k_{78} \\ 0 & 0 & 0 & k_{84} & k_{85} & k_{86} & k_{87} & k_{88} \\ 0 & 0 & 0 & 0 & 0 & k_{96} & k_{97} & k_{98} \end{bmatrix} \begin{bmatrix} \varphi_{1,z} \\ u_{2,y} \\ \varphi_{2,z} \\ u_{3,y} \\ \varphi_{3,z} \\ u_{4,y} \\ \varphi_{4,z} \\ \varphi_{5,z} \end{bmatrix}, \tag{56}
$$

where the stiffness matrix depends of the moment of inertia of the flange ($I_f$), the nominal area ($A_b$) and moment inertia ($I_b$) of the bolts, the position of the bolts $n_\alpha$, $m_\alpha$, $n_\beta$ and $m_\beta$ (see Figure 16). Additionally, the stiffness matrix has the following elements:

$k_{21} = \frac{4}{n_\alpha}$, $k_{22} = \frac{-6}{n_\alpha^2}$, $k_{23} = \frac{2}{n_\alpha}$; $k_{31} = \frac{-6}{n_\alpha^2}$, $k_{32} = \frac{12}{n_\alpha^3} + \frac{12}{m_\alpha^3} + \frac{EA_b}{L_b}$, $k_{33} = \frac{-6}{n_\alpha^2} + \frac{6}{m_\alpha^2}$,

$k_{34} = \frac{-12}{m_\alpha^3}$, $k_{35} = \frac{6}{m_\alpha^2}$; $k_{42} = \frac{2}{n_\alpha}$, $k_{43} = \frac{-6}{n_\alpha^2} + \frac{6}{m_\alpha^2}$, $k_{44} = \frac{4}{n_\alpha} + \frac{4}{m_\alpha}$, $k_{45} = \frac{-6}{m_\alpha^2}$, $k_{46} = \frac{2}{m_\alpha}$;

$k_{52} = \frac{-12}{m_\alpha^3}$, $k_{53} = \frac{-6}{n_\alpha^2}$, $k_{54} = \frac{12}{m_\alpha^3} + \frac{12}{m_\beta^3}$, $k_{55} = \frac{-6}{m_\alpha^2} + \frac{6}{m_\beta^2}$, $k_{56} = \frac{-12}{m_\beta^3}$, $k_{57} = \frac{6}{m_\beta^2}$; $k_{62} = \frac{6}{m_\alpha^2}$,

$k_{63} = \frac{2}{m_\alpha}$, $k_{64} = \frac{-6}{m_\alpha^2} + \frac{6}{m_\beta^2}$, $k_{65} = \frac{4}{m_\alpha} + \frac{4}{m_\beta}$, $k_{66} = \frac{-6}{m_\beta^2}$, $k_{67} = \frac{2}{m_\alpha}$; $k_{74} = \frac{-12}{m_\beta^3}$, $k_{75} = \frac{-6}{m_\beta^2}$,

$k_{76} = \frac{12}{m_\beta^3} + \frac{12}{n_\beta^3} + \frac{(1+\Phi)}{EI_f} \frac{EA_b}{L_b}$, $k_{77} = \frac{-6}{m_\alpha^2} + \frac{6}{n_\beta^2}$, $k_{78} = \frac{6}{n_\beta^2}$; $k_{84} = \frac{6}{m_\beta^2}$, $k_{85} = \frac{2}{m_\beta}$, $k_{86} = \frac{-6}{m_\beta^2} + \frac{6}{n_\beta^2}$,

$k_{87} = \frac{4}{m_\beta} + \frac{4}{n_\beta} + \frac{(1+\Phi)}{EI_f} \frac{EI_b}{L_b}$, $k_{88} = \frac{2}{n_\beta}$; $k_{96} = \frac{6}{n_\beta^2}$, $k_{97} = \frac{2}{n_\beta}$, $k_{98} = \frac{4}{n_\beta}$.

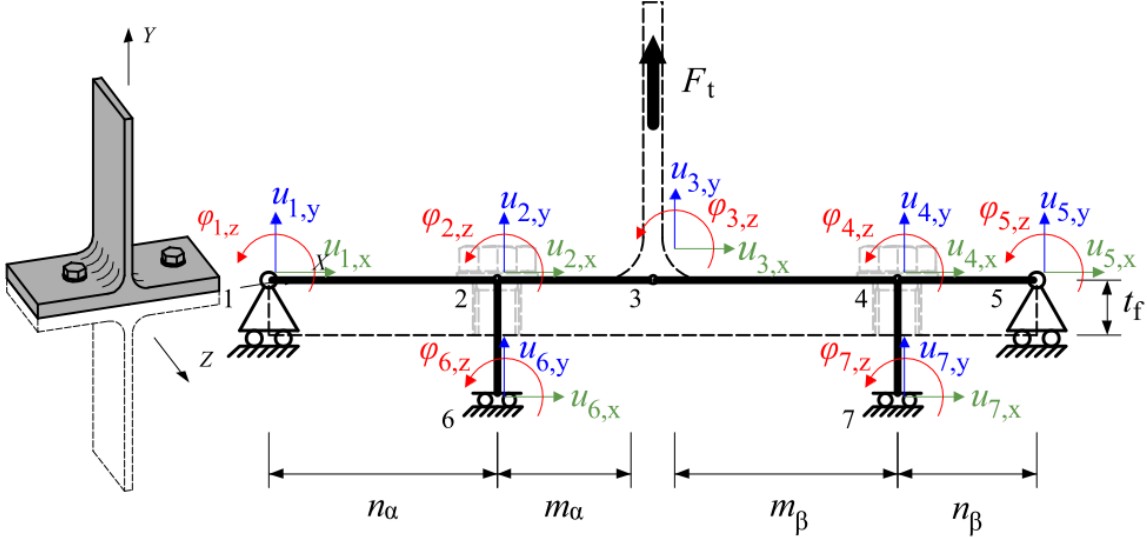

**Figure 16.** Proposed analytical model of the asymmetric T-stub [22].

Finally the equations, which were adapted from the EC3, to determine the failure modes of asymmetric T-stubs were proposed: Mode 1 Equations (1) and (2) change to Equations (57) and (58), respectively. Mode 2 Equation (3) changes to Equation (59) and the Mode 3 Equation (5).

$$
F_{Rd,1} = \frac{\varepsilon+1}{2\varepsilon} \times \frac{4M_{pl,1,Rd}}{m_\beta},
$$
$$
\varepsilon = m_\alpha / m_\beta \tag{57}
$$

$$F_{Rd,1} = \frac{A}{B} \times M_{pl,1,Rd},$$

$$A = 4(m_\alpha + m_\beta)^2 n_\alpha n_\beta - d_w \left( m_\alpha^2 (n_\alpha - n_\beta) + m_\beta^2 (n_\beta - n_\alpha) + 2m_\alpha m_\beta (n_\alpha + n_\beta) \right) \quad (58)$$

$$B = 2m_\alpha m_\beta (m_\alpha + m_\beta) n_\alpha n_\beta - d_w \left( m_\alpha m_\beta^2 n_\beta + m_\beta^2 n_\alpha n_\beta + m_\alpha^2 n_\alpha (m_\beta + n_\beta) \right)$$

$$F_{Rd,2} = M_{pl,2,Rd} \left( \frac{1}{m_\alpha + n_\alpha} + \frac{1}{m_\beta + n_\beta} \right) + B_{t,Rd} \left( \frac{1}{m_\alpha + n_\alpha} + \frac{m_\beta}{m_\alpha} \left( \frac{n_\beta}{m_\beta + n_\beta} \right) \right), \quad (59)$$

$$F_{Rd,3} = (\varepsilon + 1) B_{t,Rd}, \quad (60)$$

### 2.12. T-Stubs at Elevated Temperatures (Fire)

#### 2.12.1. EC3 Analytical Model

The behaviour of the T-stub component at high temperatures (fire) is covered by the EC3 1-2 [23] and EC3 1-8 norms [1]. The EC3 part 1–8 is modified by the strength reduction factors (SRF) of EC3 part 1-2. Equations (1)–(6) could be adapted by replacing the term $f_y$ with $K_{y,T} \times f_y$ and $B_{t,Rd}$ with $K_{b,T} \times B_{t,Rd}$. The $K_{y,T}$ and $K_{b,T}$ are found in Table 3.1 and Table D.1 of [23].

Other analytical models proposed by Spyrou [24] and Heidarpour [25] were compared with the proposed method of the Eurocode 3 when T-stubs are under transient heat transfer conditions by Gao [26] and Barata [27]. The result of the comparisons showed that the Heidarpour model provides reasonable predictions of the ultimate temperature of T-stubs, and the EC3 model predicted with a very lower accuracy the ultimate temperature. Additionally, the comparison of the deformation capacity showed that more data points should be included to improve the Spyrou and Heidarpour models.

#### 2.12.2. Spyrou's Model

Spyrou proposed a simplified model to predict the behaviour of T-stubs at ambient and elevated temperatures [24]. As for EC3, the T-stub could fail by three different mechanisms which should be modified by SRF (like EC3 for steel, for bolts Equation (61)) when T-stubs are working at elevated temperatures, and at ambient temperature for Mode 1 and Mode 2 a first plastic hinge is developed near to the flange-web interface. The behaviour of the T-stub when first plastic hinge is developed (first yielding) is determined by Equations (62)–(69):

$$
\begin{aligned}
SFR &= \quad 1, \; for \; \theta_b \leq 300 \, ^\circ C, \\
SFR &= \quad 1 - (\theta_b - 300)2.128 \times 10^{-3}, \; for \; 300 \, ^\circ C < \theta_b \leq 680 \, ^\circ C \\
SFR &= \quad 0.17 - (\theta_b - 680)5.13 \times 10^{-4}, \; for \; 680 \, ^\circ C < \theta_b \leq 1000 \, ^\circ C
\end{aligned} \quad (61)
$$

$$\delta_{cl} = \frac{FL_e^3}{48EI} - \frac{wk}{EI} \left[ \frac{L_e^3}{24} + \frac{(m + 0.5k)^3}{6} - \frac{(m + 0.5k)^2 L_e}{4} - \frac{k^2(n + 0.5k)}{24} \right], \quad (62)$$

$$\delta_{bolt} = \frac{wkL_b}{E_b A_S}, \quad (63)$$

$$\rho = \frac{\left[ \frac{(n+0.5k)L_e^2}{16} - \frac{(n+0.5k)^3}{12} \right]}{\frac{L_b EI}{E_b A_S} - \left( \frac{(n+0.5k)^3}{6} + \frac{k^2(n+0.5k)}{24} + \frac{(n+0.5k)(m+0.5k)^2}{2} - \frac{(n+0.5k)L_e^2}{8} - \frac{k^3}{384} \right)}, \quad (64)$$

$$M_p = \frac{\left( 2L_{eff} \right) t_f^2 f_y}{4}, \quad (65)$$

$$F_{cl,pl} = \frac{2 \left( wk(n + 0.5k) - M_p \right)}{n + k + m}, \quad (66)$$

$$F_{bl,pl} = \frac{2 \left( wk(n + 0.5k) - wk(0.125k) - M_p \right)}{n + 0.5k}, \quad (67)$$

$$F_{bolt,pl} = \frac{2A_s f_{by}}{\rho}, \tag{68}$$

$$F_{1st} = \min\left(F_{cl,pl}, F_{bl,pl}, F_{bolt,pl}\right), \tag{69}$$

(1) Mode 1: after the first yielding occurs in the T-stub flange, the yielding and fracture of the bolts is observed. The required force—the prying force; bolt force and the displacement—after the first yielding (located at the bolt), are determined by Equations (70)–(73). After the yielding of the bolt, the bolts take any increase of the T-stub force ($\Delta wk^I_{bolt,pl}$) until fracture, the prying force ($\Delta Q$) cannot increase more and the incremental deflections ($\Delta \delta^I_{cl}$) are calculated by Equations (74)–(76):

$$F^I_{bolt,pl} = (F_{1st} + \Delta F) = \frac{2M_p + 4A_s f_{by}\left(2 + \frac{k}{2}\right)}{n + k + m}, \tag{70}$$

$$wk^I_{bolt,pl} = (wk + \Delta wk) = \frac{0.5 F^I_{bolt,pl}(n + k + m) - M_P}{n + 0.5k}, \tag{71}$$

$$Q^I = (Q + \Delta Q) = \frac{0.5 F^I_{bolt,pl}(m + 0.5k) - M_P}{n + 0.5k}, \tag{72}$$

$$\delta^I_{cl} = (\delta_{cl} + \Delta \delta_{cl}) = \frac{wk^I_{bolt,pl}}{EI}\left[\frac{(n+k+m)^3}{6} - \frac{(0.5k+m)^3}{6}\right] - \frac{F^I_{bolt,pl}}{2EI}\frac{(n+k+m)^3}{6} + \frac{A(n+k+m)+B}{EI},$$
$$A = wk^I_{bolt,pl}\left[\frac{k^3}{384(n+0.5k)} - \frac{(n+0.5k)^2}{6} - \frac{k^2}{24}\right] + \frac{F^I_{bolt,pl}}{2}\frac{(n+0.5k)^2}{6} + \frac{EI\delta_{bl}}{n+0.5k} + \frac{\Delta wkEIL_b}{E_b A_s(n+0.5k)} \tag{73}$$
$$B = wk^I_{bolt,pl}\left(\frac{k^2(n+0.5k)}{24}\right)$$

$$\Delta wk^I_{bolt,pl} = \frac{\Delta F}{2} = 2A_s f_{bu} - wk^I_{bolt,pl}, \tag{74}$$

$$\Delta Q = 0, \tag{75}$$

$$\Delta \delta^I_{cl} = \frac{\Delta F}{EI}\left[\frac{(m+0.25k)^2(m+0.5k)}{8} - \frac{(m+0.25k)^3}{24} + \frac{k^3}{1536} + \frac{EIL_b}{2E_{tb}A_s}\right], \tag{76}$$

(2) Mode 2: after the first plastic hinge (Equations (22)–(69)), a second hinge is developed in the T-stub flange and the T-stub total force ($F^{II}_{bl,pl}$), the total bolt force ($wk^{II}_{bl,pl}$) and the total prying force ($Q^{II}_{bl,pl}$) can be calculated by Equations (77)–(82). In the zone until the yielding of the bolt and between the yielding and fracture of the bolt the same parameters are calculated by Equations (83)–(88) and by Equations (89)–(93):

$$F^{II}_{bl,pl} = (F_{1st} + \Delta F) = \frac{2M_p\left(2n + \frac{7k}{8}\right)}{mn + 0.375k(m + n) + 0.125k^2}, \tag{77}$$

$$\Delta wk = 0.5\Delta F\left[\frac{m + 0.5k}{n + 0.5k} + 1\right], \tag{78}$$

$$\Delta Q = 0.5\Delta F\left[\frac{m + 0.5k}{n + 0.5k}\right], \tag{79}$$

$$wk^{II}_{bl,pl} = wk + \Delta wk, \tag{80}$$

$$Q^{II}_{bl,pl} = Q + \Delta Q, \tag{81}$$

$$\Delta\delta_{cl} = \frac{\Delta Q}{EI}\left[\frac{(n+k+m)^3}{6}\right] - \frac{\Delta wk}{EI}\frac{[m+0.5k]^3}{6} + \frac{C(n+k+m)}{EI} + \frac{D}{EI},$$

$$C = \Delta wk\left[\frac{k^3}{384(n+0.5k)} - \frac{k^2}{24} + \frac{EIL_b}{E_b A_s(n+0.5k)}\right] - \frac{\Delta Q[n+0.5k]^2}{6} \tag{82}$$

$$D = \Delta wk\left[\frac{k^2(n+0.5k)}{24}\right]$$

$$F^{II}{}_{bolt,pl} = \left(F^{II}{}_{bl,pl} + \Delta F\right), \tag{83}$$

$$\Delta wk = 0.5\Delta F = 2A_s f_{by} - wk^{II}{}_{bl,pl}, \tag{84}$$

$$\Delta Q = 0, \tag{85}$$

$$wk^{II}{}_{bl,pl} = wk + \Delta wk, \tag{86}$$

$$wk^{II}{}_{bolt,pl} = wk^{II}{}_{bl,pl} + \Delta wk, \tag{87}$$

$$\Delta\delta_{cl} = \frac{\Delta F}{E_t I}\left[\frac{(n+0.25k)^2(m+0.5k)}{8} - \frac{(m+0.25k)^3}{24} + \frac{k^3}{1536} + \frac{E_t IL_b}{2E_b A_s}\right]$$

$$E_t = 1.5\%E \tag{88}$$

$$F^{II}{}_{bolt,ul} = (F^{II}{}_{bolt,pl} + \Delta F), \tag{89}$$

$$\Delta wk = 0.5\Delta F = 2A_s f_{bu} - wk^{II}{}_{bolt,pl}, \tag{90}$$

$$\Delta Q = 0, \tag{91}$$

$$wk^{II}{}_{bolt,ult} = wk^{II}{}_{bolt,pl} + \Delta wk, \tag{92}$$

$$\Delta\delta_{cl} = \frac{\Delta F}{E_t I}\left[\frac{(n+0.25k)^2(m+0.5k)}{8} - \frac{(m+0.25k)^3}{24} + \frac{k^3}{1536} + \frac{E_t IL_b}{2E_{tb} A_s}\right],$$

$$E_{tb} = 1\%E_b \tag{93}$$

(3)   Mode 3: T-stubs remains elastic and plastic fracture of the bolts occurs.

In this mode the force that is required for the bolt to yield is determined by assuming $F_{1st} = F_{bolt,pl}$ in Equation (69). Then the increment necessary to fracture the bolts is determined by Equation (90), but $wk^{II}{}_{bolt,pl}$ is changed for $2A_s f_{by}$. The displacement is given by Equation (93).

### 2.12.3. Heidarpour's Model

Heidarpour proposed a model based on the observation that depending on the ratio of flexural stiffness of the end plate which acts in series with the column flange to the axial stiffness of the bolts to evaluate the thermo-elastic and plastic behaviour of flexible T-stub [25]. In this analytical model, the degradation of the components material due to thermal loads is considered. The analysis to determine the behaviour of the T-stubs starts by calculating the required force ($F_y^1$) and the deflection of the T-stub ($\delta_y^1$) when the first yield point is formed, see Equations (94)–(98) when $Q > 0$ and Equations (99)–(102) when $Q = 0$:

$$F_y^1 = \min(F_f, F_{bl}, F_b), \tag{94}$$

$$F_f = \eta_{Ys}\left(\frac{\frac{\eta_{Es}}{n_b\eta_{Eb}}+\beta\frac{K_{b0}}{\overline{K_0}}}{(e+m)\frac{\eta_{Es}}{n_b\eta_{Eb}}+e(\beta-\alpha)\frac{K_{b0}}{\overline{K_0}}+m\beta\frac{K_{b0}}{\overline{K_0}}}\right) \times \min\left(M_{ep0}, M_{cp0}\right),$$

$$K_{b0} = \frac{E_{b0}A_s}{l_b}$$

$$\overline{K_0} = \frac{K_{e0}+K_{c0}}{K_{e0}K_{c0}} \tag{95}$$

$$F_{bl} = \eta_{Ys}\left(\frac{\frac{\eta_{Es}}{n_b\eta_{Eb}} + \beta\frac{K_{b0}}{\overline{K_0}}}{e(\alpha - \beta)\frac{K_{b0}}{\overline{K_0}} - \frac{e\eta_{Es}}{n_b\eta_{Eb}}}\right) \times \min\left(M_{ep0}, M_{cp0}\right), \tag{96}$$

$$F_b = \eta_{Yb}\left(\frac{1}{\alpha} \times \frac{\eta_{Es}}{\eta_{Eb}} \times \frac{\overline{K_0}}{K_{b0}} + \frac{n_b \beta}{\alpha}\right)B_{y0}, \tag{97}$$

$$\delta_y^1 = \frac{F_y^1}{24\eta_{Es}\overline{K_0}}\left(1 - \frac{24\alpha^2}{\frac{\eta_{Es}\overline{K_0}}{n_b\eta_{Eb}K_{b0}} + \beta}\right), \tag{98}$$

$$F_y^1 = \min(F_f, F_b), \tag{99}$$

$$F_f = \eta_{Ys} \times \frac{\min(M_{ep0}, M_{cp0})}{m}, \tag{100}$$

$$F_b = n_b\eta_{Yb}B_{y0}, \tag{101}$$

$$\delta_y^1 = F_y^1\left(\frac{1}{n_b\eta_{Eb}K_{b0}} + \frac{m^3}{3L^3\eta_{Es}\overline{K_0}}\right), \tag{102}$$

where $n_b$ is the number of bolts at each bolt line, $\eta_{Es} = E_{sT}/E_{s0}$, $\eta_{Eb} = E_{bT}/E_{b0}$ are the ratio between the Young's modulus at an elevated temperature and at ambient temperature of the steel and bolt, respectively, $\beta = 0.5l^2 - 2/3l^3$, $\alpha = 0.125l - 1/6l^3$, $l = e/L = e/(2(m+e))$. The terms $e$ and $m$ are shown in Figure 4. In Equation (95) $K_{b0}$ is the stiffness of the bolt at ambient temperature and $A_s$ and $l_b$ are the effective area and length of bolt; $\overline{K_0}$ is the equivalent stiffness of the endplate ($K_{e0}$) and column ($K_{c0}$) at ambient conditions. $\eta_{Ys} = f_{ysT}/f_{ys0}$, $\eta_{Yb} = f_{ybT}/f_{yb0}$, $M_{ep0} = f_{ye0} \times S_e$, $M_{cp0} = f_{yc0} \times S_c$. The terms $f_{ysT}$, $f_{ybT}$ and $f_{ys0}$, $f_{yb0}$ are the yielding stress of the steel and bolt at an elevated temperature and the yielding stress of the steel and bolt at ambient temperature respectively. The terms $f_{ye0}$, $f_{yc0}$ and $S_e$, $S_c$ are the yielding stress at ambient temperature of the endplate and column flange and the plastic section modulus of the endplate and column flange, respectively.

After the first plastic hinge has appeared a second plastic hinge is formed and three possibilities could be developed for the analysis: Case 1, Case 2 and Case 3.

(1)  Case 1: when the first plastic hinge occurs at the fillet of the end plate or column flange, the bending moment is equal to $M_{epT}$ ($\eta_{Ys} \times M_{ep0}$) or equal to $M_{cpT}$ ($\eta_{Ys} \times M_{cp0}$). Therefore, the second yield point could be formed at the bolt line or at the bolt. The force developed at the second yield point is calculated by Equations (103)–(109):

$$F_y^2 = F_y^1 + \Delta F^1, \tag{103}$$

$$\Delta F^1 = \min(\Delta F_{bl}, \Delta F_b), \tag{104}$$

$$\Delta F_{bl} = \frac{(\min(M_{epT}, M_{cpT}) - eQ_y^1}{m}, \tag{105}$$

$$\Delta F_b = \frac{e(n_b\eta_{Yb}B_{y0} - B_y^1)}{m+e} \tag{106}$$

$B_y^1$ and $Q_y^1$ are calculated by replacing $F_y^1$ in Equations (107) and (108), respectively:

$$B = \left(\frac{\alpha}{\frac{\eta_{Es}\overline{K_0}}{n_b\eta_{Eb}K_{b0}} + \beta}\right)F \tag{107}$$

$$Q = B - F \tag{108}$$

$$\delta_y^2 = \delta_y^1 + \Delta F^1\left(\frac{2m^2(e+m)}{3L^3\eta_{Es}\overline{K_0}} + \left(1 + \frac{m}{e}\right)^2\frac{1}{n_b\eta_{Eb}K_{b0}}\right) \tag{109}$$

When $\Delta F^1 = \Delta F_{bl}$ and $M_{epT} = M_{cpT}$, the second and third plastic hinges are formed simultaneously and the T-stubs behave as a mechanism. Therefore, no further load can be carried by the T-stub and the ultimate value of the force ($F_u$) and deflection ($\delta_u$) can be calculated by Equations (103) and (109). However, when $M_{epT} > M_{cpT}$ (or $M_{epT} < M_{cpT}$),

the force can increase until the bolts start to yield and after a while to fracture. The load and deflection increments which cause the yielding at the bolts are calculated by Equations (110)–(113)

$$\Delta F^2 = n_b \eta_{Yb} B_{y0} - B_y^2, \tag{110}$$

$$\delta_y^3 = \delta_y^2 + \Delta F^2 \left( \frac{1}{n_b \eta_{Eb} K_{b0}} + \frac{m^3}{3L^3 \eta_{Es} \overline{K_{t0}}} \right), \tag{111}$$

$$B_y^2 = B_y^1 + \Delta F_{bl} \left( \frac{m}{e} + 1 \right), \tag{112}$$

$$\frac{1}{\overline{K_{t0}}} = \frac{L^3}{E_{ts0}} \left( \frac{1}{I_c} + \frac{\eta_{Ets} E_{ts0}}{n_{Es} E_{s0}} \times \frac{1}{I_e} \right), \tag{113}$$

where $\eta_{Ets}$ is the retention ratio of the tangent modulus of structural steel at high temperature to that at ambient temperature, see Equation (114):

$$\eta_{Ets} = \frac{E_{tsT}}{E_{ts0}}, \tag{114}$$

Finally, the fracture of the bolt when $\Delta F^1 = \Delta F_{bl}$ and $M_{epT} > M_{cpT}$ (or $M_{epT} < M_{cpT}$) the next load and deflection increments provoke the fracture of the bolt. The ultimate tensile force resistance and deflection are determined by Equations (115)–(118)

$$\Delta F^3 = n_b \left( \eta_{Ub} B_{U0} - \eta_{Yb} B_{y0} \right), \tag{115}$$

$$\delta_u = \delta_y^3 + \Delta F^3 \left( \frac{1}{n_b \eta_{Etb} K_{tb0}} + \frac{m^3}{3L^3 \eta_{Ets} \overline{K_0}} \right), \tag{116}$$

$$F_u = F_y^2 + \Delta F_y^2 + \Delta F^3, \tag{117}$$

$$K_{tb0} = \frac{E_{tb0} A_s}{l_b}, \tag{118}$$

where $\eta_{Ub}$ and $\eta_{Etb}$ are the retention ratios of the ultimate tensile strength and tangent modulus of the bolts at elevated temperature to those at ambient temperature.

$$\eta_{Ub} = \frac{f_{ubT}}{f_{ub0}}, \tag{119}$$

$$\eta_{Etb} = \frac{E_{tbT}}{E_{tb0}}, \tag{120}$$

On the other hand, when $\Delta F^1 = \Delta F_b$ the bolt yield and the bolt yield and load can increase until the bolt fractured. The load increment is calculated by Equation (121) and the ultimate values of force and deflection are determined by Equations (122) and (123):

$$\Delta F^2 = n_b \left( \eta_{Ub} B_{U0} - \eta_{Yb} B_{y0} \right), \tag{121}$$

$$F_u = F_y^2 + \Delta F^2, \tag{122}$$

$$\delta_u = \delta_y^2 + \Delta F^2 \left( \frac{1}{n_b \eta_{Etb} K_{tb0}} + \frac{m^3}{3L^3 \eta_{Es} \overline{K_0}} \right), \tag{123}$$

(2)   Case 2: in this case the first yielding is produced at the bolt line, and the prying force cannot be increased more. Equation (104) becomes Equation (124):

$$\Delta F^1 = \min \left( \Delta F_f, \Delta F_b \right), \tag{124}$$

$$\Delta F_f = \frac{\min \left( M_{epT}, M_{cpT} \right)}{m}, \tag{125}$$

$$\Delta F_b = n_b \eta_{Yb} B_{y0} - B_y^1, \tag{126}$$

$$\delta_y^2 = \delta_y^1 + \Delta F^1 \left( \frac{1}{n_b \eta_{Eb} K_{b0}} + \frac{m^3}{3L^3 \eta_{Es} \overline{K_0}} \right), \tag{127}$$

When $\Delta F^1 = \Delta F_f$ and $M_{epT} = M_{cpT}$ the assembly reaches mechanism status and the $F_u = F_y^1 + \Delta F^1$ and $\delta_u = \delta_y^2$. Nevertheless, when $M_{epT} > M_{cpT}$ (or $M_{epT} < M_{cpT}$) the force can be increased until the bolts yield and fracture. The corresponding load and deflection increments can be calculated by employing Equations (110), (111) and (115)–(117), where in Equation (110) $B_y^2 = B_y^1 + \Delta F_f$.

Under other conditions, when $\Delta F^1 = \Delta F_b$, two possibilities could be developed. First a third plastic hinge could be formed at the fillet. The second possibility is when the bolt fracture before any plastic hinge appears at the fillet. In former scenario, when $M_{epT} = M_{cpT}$ the T-stub behaves as a mechanism, whilst when $M_{epT} > M_{cpT}$ (or $M_{epT} < M_{cpT}$) the applied load and deflections can increase until the bolt fracture and are determined by Equations (115)–(117). Nevertheless, in the latter scenario, the second increment of load is calculated by Equation (128), where $\Delta F_b^2$ is obtained in Equation (121). Additionally, Equations (122) and (123) gives $F_u$ and $\delta_u$:

$$\Delta F^1 = \Delta F_b^2, \tag{128}$$

(3) Case 3: in this case the bolts yield first, and the prying forces are equal to zero. Therefore, any increment in the load is carried by the bolts until the bolt fractures. According to this, Equation (104) changes to Equation (129), where $\Delta F_b^1$ is obtained in Equation (121). Finally, the ultimate central separation and the tensile resistance of the assembly is given in Equations (129)–(131)

$$\Delta F^1 = \Delta F_b^1, \tag{129}$$

$$\delta_u^2 = \delta_y^1 + \Delta F^1 \left( \frac{1}{n_b \eta_{Etb} K_{tb0}} + \frac{m^3}{3L^3 \eta_{Es} \overline{K_0}} \right), \tag{130}$$

$$F_u = F_y^1 + \Delta F^1, \tag{131}$$

## 3. Experimental Testing

Experimental tests obtain the most reliable and accurate information of the T-stub component behaviour [2]. However, developing the experimental tests for typical routine designs is expensive. Therefore, the experimental results are mostly reserved for research schemes [28]. Appendix A contains relevant information of the T-stubs studied by some of the authors that are presented in this review.

In 1965, Douty and McGuire [3] did various monotonic experimental tests of the three most common moment connections to study their performance, design, and use on plastically designed structures. Although the investigation studied three different configurations, the T-stub connection was more studied than the end plate connection.

This research work divided the investigation into five main parts:

(1) The T-stub web-to-beam flange.
(2) T-stub to column connection.
(3) Assembled T-stub connection, end plate connection.
(4) End plate connection
(5) Suggested a method for the design (semi-empirical model).

Part 2 studied the T-stub flange in the tension zone and the prying force's effect on the tension bolt. As a result of this, the authors concluded that the prying force increased the tension bolt force and identified the importance of the material strain hardening, as well [3].

In 1974, Nair et al. [29] studied by experimental testing the bolts that are subject to tension and prying actions. The tests that were performed were divided into three: tests

of single bolt connections, static tests of T-connections and fatigue tests of T-connections. The tests of single bolts connections were performed by applying the tension load in several cycles. The static tests of T-connections were conducted on two types of connection which were designated by the letters U (only the ultimate load was measured) and S (the behaviour of the connection under lower levels of load was also studied). The fatigue tests of the T-stub were conducted, and the failure of the connection was defined as the complete fracture of one or more bolts. However, if the failure had not occurred until the number of test cycles was greater than 3,000,000, the experiment was terminated.

Also in 1974, Zoetemeijer [30] performed a series of tests at the Stevin Laboratory of the Delft University of Technology in The Netherlands to validate a design method for the tension side of statically loaded, bolted beam-to-column joints (T-stub connection and end-plate connection). In this research, four tests were executed to study the T-stub connection, see Figure 17a, and the failure modes were divided into two mechanisms (mechanism A and B, respectively). Mechanism A is governed by the bolt fracture, and two possibilities can develop. One is produced when the bolt fails, and the T-stub flange does not reach the yielding stress (it is equivalent to the EC failure mode 3), and the other is characterised by some yielding in the T-stub flange and the failure of the bolt (similar to the EC failure mode 2). Mechanism B is characterised by the yielding of the flange, and the prying force reaches the maximum value (similar to EC failure mode 1).

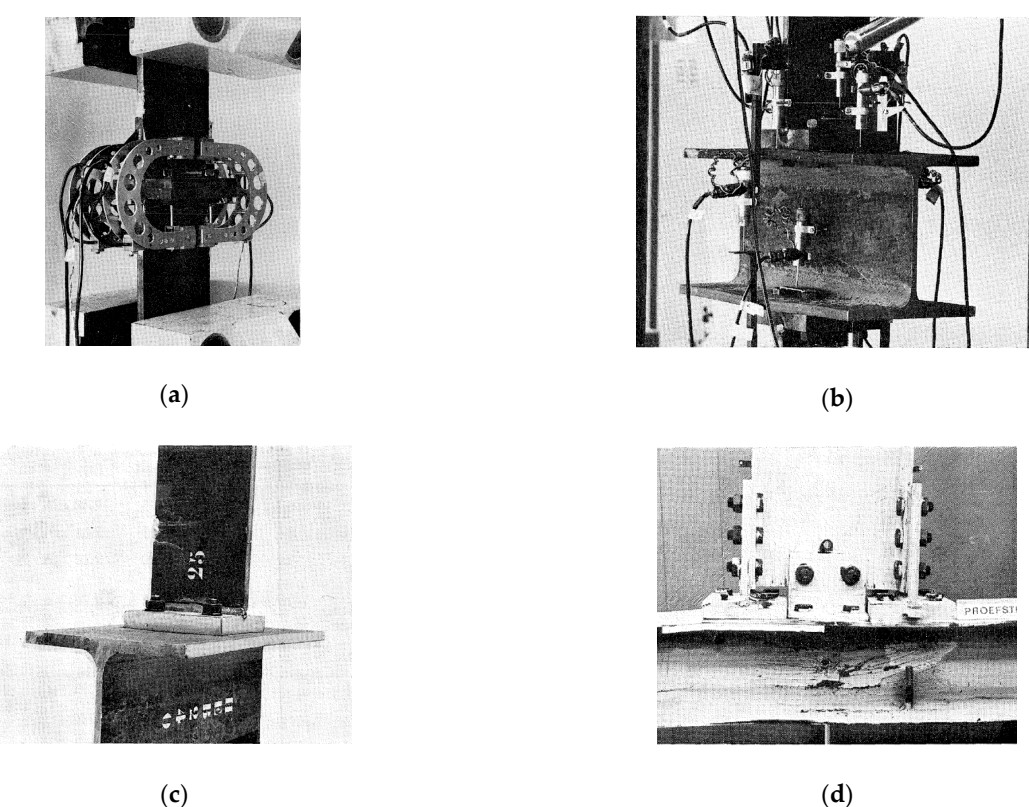

(**a**)　　　　　　　　　　　　　　　　　　(**b**)

(**c**)　　　　　　　　　　　　　　　　　　(**d**)

**Figure 17.** (**a**) T-stub connection test; (**b**) test to check the effective length; (**c**) test to check the philosophy of T-stub on to column flange; (**d**) test to check the serviceability [30].

On the other hand, the tests performed to study the T-stub flange to the column connection were divided into two groups. The first group consisted of nineteen tests that were executed to check the theory of the effective length, see Figure 17b. The second group consisted of five tests to check the philosophy that the design method of T-stubs can be applied to analyse the column flange, see Figure 17c. Furthermore, the research was focused on determining the serviceability limits for the design method. Therefore,

twenty-three whole joints were tested (seventeen T-stub connections and six end-plate connection), see Figure 17d.

In 1976, Agerskov [31] proposed an analytical method, see Figure 18a, to determine the yield force and the bolt force for T-connections and end plate connections. This method takes into consideration the effect of the prying force, which increases the bolt force. Therefore, Aggerskov performed four experimental tests on welded T-connections and fifteen tests on beam-to-beam end plate connections to validate this theory. The experimental tests showed that the mechanism model that considered the formation of plastic hinges at the fillet toe and the bolt line were unlikely to appear. This mechanism model was taken into account by Douty, Kato, Nair, etc. Furthermore, this research work clarified that end plate bolted connection could be designed as a T-connection and two different manners of calculate the conventional bolt length $L_b$, see Figure 18b, for snug-tightened bolts and for preloaded bolts (bolt stiffness).

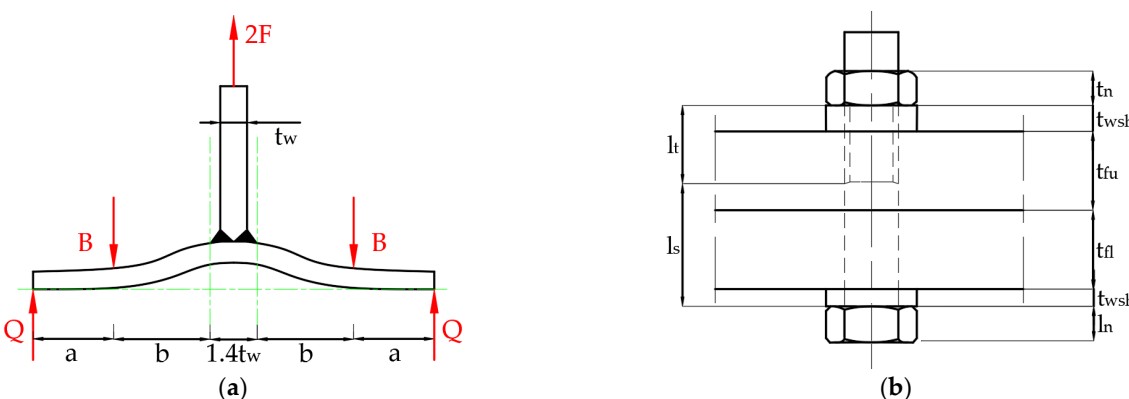

**Figure 18.** (**a**) Analytical model of connection; (**b**) Model of conventional bolt length [31].

In 1977, Packer and Morris [32] idealized the tension zone of the extended end plate beam and the column flange as an isolated T-stub. The failure mode was studied according to the yield line method, but they allowed curved yield boundaries, which accurately predicted the flexural yield loads in unstiffened and normally stiffened connection. They defined the failure modes by three mechanisms: Mechanism A (EC Failure mode 3), Mechanism B (EC Failure mode 2) and Mechanism C (EC Failure mode 1).

In 1983, Zoetemeijer continued his research on end-plate moment connections and published a proposal for the standardization of the design formulas of extended end-plate connection [33], which are based on the results of experimental tests. This standardization concluded that connections with extended end plates develop enough rotation capacity for plastic design of a beam if the connection is design according to the plastic design formulas of the standardization (see the Table 3.2 of [33]), the beam-span is limited to 30 times the beam depth and the beam span is limited to 16 m.

In 1998, Faella et al. [34] studied the influence of bolt preloading on T-stubs. These T-stubs were obtained from laminated profile HEA and HEB, steel grade Fe430, and were connected by high strength bolts 10.9 with diameter of 20 mm (six specimens) and 12 mm (10 specimens). The bolt preloading was applied in three different levels: the first level was snug tight, the second level corresponded to 40% (275 Nm) of the bolt yield stress, for diameter of 20 mm, and to a 60% of the bolt yield stress (85 Nm), for a diameter of 12 mm. The third level corresponded to an 80% of the bolt yield resistance equal to 550 Nm and 113 Nm for bolt diameter 20 mm and 12 mm, respectively. It was observed that the bolt preloading significantly affects the stiffness of bolted joints. Moreover, the reliability of predicting the T-stub stiffness was studied by the component method (non-preloaded and preloaded).

In 1999, Swanson described 48 T-stub tests in [35]. These experimental tests were performed to develop design rules for T-stub connections which would result in a full

strength connection, ductile behaviour, and connection stiffness. In this research Swanson was focused to study the T-stub component, see Figure 19, subject to tensile static load (studied by Douty and Mc Guire, and Aggerskov), and cycle loads. The results about the comparison between monotonically tested T-stubs and cyclically tested T-stubs showed that the monotonic test data provided an accurate envelope of the cycle results.

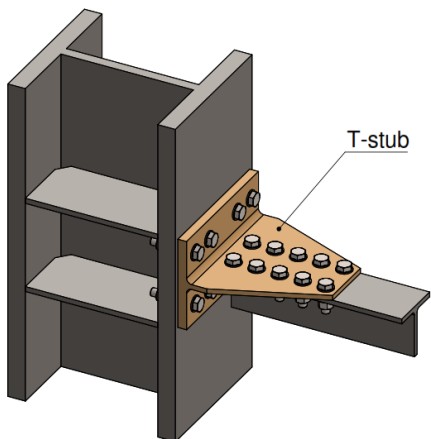

**Figure 19.** Typical T-stub configuration tested [35].

In 2001, Piluso et al. [36] presented the results of an experimental program that was devoted to validating a theoretical model, which was proposed in a previous investigation [37]. The experimental program tested 12 T-stub to evaluate the plastic supply. The specimens were characterised by different values of the ratio between the flexural resistance of the flanges and the axial resistance of the bolts (notice that the flexural resistance of the bolt was not taken into account). The results were compared with the theoretical model with the force displacement curve. This comparison reveals a satisfactory degree of accuracy and, as a consequence, constitutes the validation of the proposed theoretical model.

In 2001, Spyrou and Davison [38] studied the behaviour of T-stub connections that work at high temperatures due to the fact the ability of the joints to sustain loadings is considerably impaired [39]. The tests employed a furnace, where the T-stubs were heated at a temperature above 300 °C and under 800 °C, and then tested. The displacement of the specimens was taken with a solid-state charge-coupled device (CCD) camera. Even though when the investigation was focused on the suitability of the CCD measurements, the results of the tests showed that the ultimate resistance at 600 °C decreased more than 56% of the resistance at ambient temperature.

In 2004, Girão et al. [40] carried out 32 tests on T-stub bolted connections made up of welded plates at the Delft University of Technology. The investigation provides insight into the behaviour of different types of assemblies in terms of resistance, stiffness, deformation capacity and failure modes. The variables that were taken into account were the following:

- The weld throat thickness.
- The size of T-stub.
- The steel grade.
- The presence of transverse stiffness and the T-stub orientation.

These tests unveiled that the welding procedure is essential to develop a ductile behaviour in the connection. The major contributions of the overall T-stub deformation are the deformation of the flange and the bolt deformation. Most of the time, the maximum deformation is reached when the T-stub flange cracking happens before the bolt cracking. However, flange cracking could happen before the connection develops its theoretical deformation because the welding procedure could change the microstructure of the flange in the HAZ. Therefore, the mechanical properties could decrease by 20%. Finally, it was concluded that the thickness of the throat affects the stiffness and the resistance of the

connection in a direct proportion. The effect of the width affects the stiffness and the resistance directly as well. Higher bolt diameter increases the bolt resistance and enhances the resistance, stiffness and ductility of the connection; identical T-stubs yield higher resistance and lower deformation capacity for higher steel grades. The stiffener decreases the deformation capacity. Moreover, for stiffened T-stub, the orientation of the elements is not relevant at the stiffener side; in the case of unstiffened T-stub, the two plates become in contact when a tensile load is applied to the connection.

In 2008, Piluso et al. [41] performed a series of experimental testing at the Material and Structure Laboratory of the Department of Civil Engineering of Salerno University to validate a new refined theoretical model of an earlier (2001) model that was mentioned above [37]. Because of the lack of precision of the original theoretical model to predict the plastic deformation capacity according to failure mode 2, some improvements were implemented in the original model, such as the bolt preloading and the effects of bending on the bolts were not disregarded. The experimental results were compared with the two theoretical models with the force-displacement curve and showed that the refined model's ultimate force coincided with the experimental value. However, the ultimate displacement was not the same due to bolt ultimate strain was taken as the lower value provided by the manufacturer.

In 2012, Carazo published his doctoral thesis [42], which was focused on study of the behaviour of T-stub connection by using optical methods to measure the stresses and strains in the elements by three different manners: digital image correlation (DIC), thermo-elasticity, and photo-elasticity. The results of the measures validate the Eurocodes prescription, and also were used to calibrate and validate a finite element model, which was developed for the research. However, the results of the tests showed that the simplifications of the T-stub in the EC do not reproduce the behaviour well, in some cases. The behaviour of the T-stub is well represented in the elastic zone, which is defined by the resistance and stiffness, but in the plastic zone the deviation of the results is higher.

In 2016, Jiménez de Cisneros et al. [22,43] started studies on asymmetrical T-stub components by testing two types of T-stubs at the Civil Engineering Department of Coimbra University (Portugal). The experimental program tested a total of two T-stubs which were symmetrical and asymmetrical as well. Furthermore, T-stubs were designed to fail according to the failure mode 3 of the Eurocode [1] since the utmost characteristics that were pretended to be observed was the asymmetrical distribution of bolt load and the premature bolt failure due to the asymmetry. The experimental testing results were used to calibrate a finite element model and an analytical model. The results also demonstrated that the asymmetric T-stubs fail for lower loads than the symmetric T-stubs because of the load distribution on the bolts.

In 2016, Wang et al. [44] tested 69 T-stub connected to a hollow structural section (HSS) under a cyclic load to characterise the low cycle fatigue response. The specimens were adequately designed to eliminate the flange's bending ($t_f$ = 25 mm), the brittle fracture of the weld and left only two sources of plastic deformation: the flexibility of the HSS column profile and the bolts. The tests were performed in a fatigue testing machine according to the cyclic testing guidance of ATC-24. Therefore, the concept of "multi-specimens programme" was used, and the cyclic load was applied in displacement control utilising constant amplitude sinusoidal cycles with a frequency of 0.35 Hz. The characteristics of the connections under cyclic load were evaluated and compared in terms of typical failures modes (failure mode (i) flexural yielding of HSS column face without bolt failure) and failure mode (ii) (moderate local yielding of HSS column wall and bolt fracture), hysteretic load-deformation relation (failure mode (ii) was prone to exhibit a pinching manner in its hysteretic load-deformation curve), and degradation of strength and energy dissipation capacity. Furthermore, the low cyclic fatigue life was analysed according to Eurocode 3 part 1.9 [45] and then reanalysed by taking into account the geometric details of the connection. Finally, a proposed model, which considered the stiffness and energy

dissipation, was compared with the commonly used damage models, and it was showed that the new model could give a reasonable prediction of experimental evidence.

In 2016, Cabaleiro et al. [14] studied the behaviour of T-stubs connected with clamps. For this purpose three specimens were fabricated from IPE220 (S235) and connected to a rigid base through two clamps by means of two high strength bolts (class 8.8). The specimens were tested under a monotonic load, which vary from 0 to 40 kN. In these tests the sizes of the front levers of the clamps were for "n" = 5, 15 and 25 mm, with a rear level "b" of 16.5 mm, see Figure 11a to see the geometry. These experimental tests were performed to validate an analytical model and to calibrate a FEM model. The results of the tests showed that with the n = 5, the deformation and failure of the specimen is produced by a semi-rigid failure mode, in the case of n = 15 the failure is produced in a combined form, while for n = 25 failure is produced by failure of the bolts.

In 2017, Zhu et al. [18] carried out tests to study the one-side bolted T-stub through thread holes under tension strengthen with backing plates, and to propose a design method. These tests were performed in a universal servo-controlled hydraulic testing machine. The specimens used M20 8.8 bolts, the flange thickness varying from 6 to 20 mm, the backing plates had a thickness of 10 mm, and the T-stubs were connected to a T-stub base with a flange thickness of 30 mm and web thickness of 20 mm to ensure the elastic zone in the base. In this research, two new failure modes were proposed to might occur, and they should be investigated: (1) Mode 4 the hole thread failure, see Figure 20a, and (2) Mode 5, the T-stub flange yielding accompanied with hole thread failure, see Figure 20b. The results of the tests showed that two failure modes were observed during the test with the change of the flange thickness, which were Mode 1 (according to EC3) and Mode 5. Apparently, the backing plate increased the yield strength of the flange, see Figure 20c. However, the effect of increasing the flange thickness is of the utmost relevance, Figure 20c. Furthermore, The T-stub with a flange thickness of 16 mm is more ductile than the T-stub of flange thickness of 6 mm and backing plate of 10 mm, see Figure 20c.

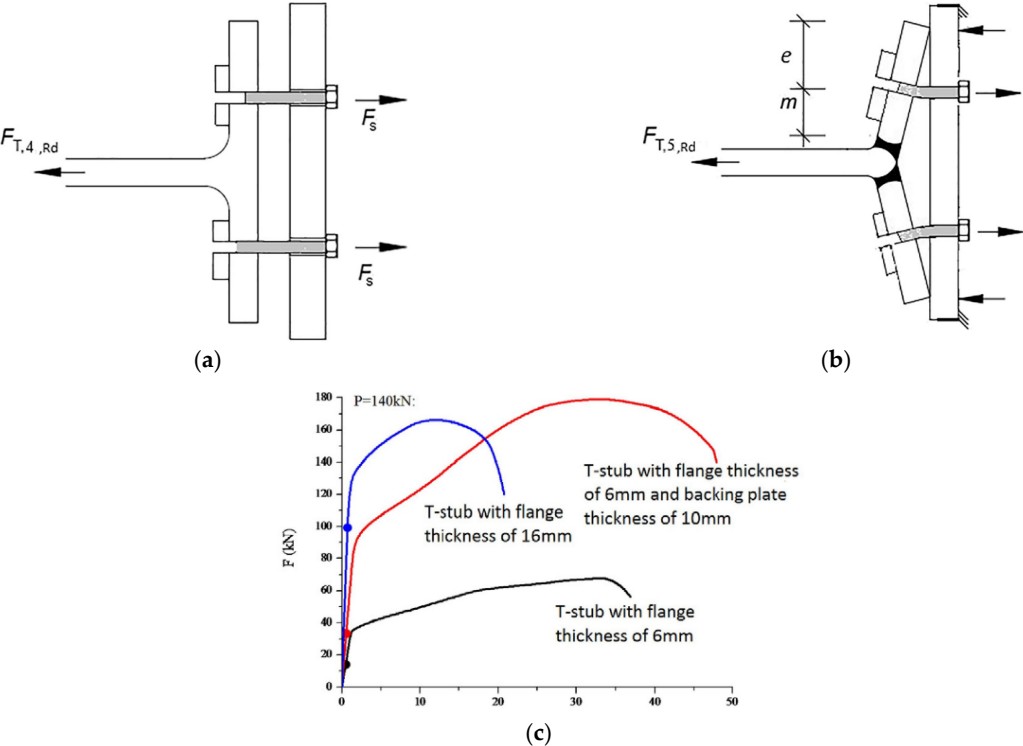

**Figure 20.** (**a**) Failure Mode 4; (**b**) Failure Mode 5; (**c**) T-stub with same flange thickness of 6 mm or thread length of 16 mm [18]. "Reproduced with permission from Luciano M. Bezerra, Jorge Bonilla, Wellington A. Silva, William T. Matias, Engineering Structures; published by Elsevier, 2017".

In 2019, Bao et al. [46] studied the mechanical behaviour of bolts used to connect T-stubs. Six T-stubs made of Q235B steel were connected with grade 10.9 bolts to a base plate (T-stub with a quite thick flange) and tested while strain gauges acquired the data of the behaviour of the bolts. The experiments showed that the bending stress reached a value ranging from 13% to 45% of the total tensile stress when the bolts yielded. Therefore, it is not advisable to disregard the effects of bending moments in the design. Furthermore, the authors observed that the increment of the flange thickness change the failure mode from mode 1 to mode 2 (similar of EC3 failure modes). It should be mention that the data that was obtained from the experimental part was then used to calibrate the finite element model developed for this investigation.

In 2020, Bezerra et al. [47] studied the behaviour of T-stub connections connected to a rigid base, see Figure 21. The researchers carried out nine T-stub tests in the Laboratory of Structures at the University of Brasilia. These T-stubs were tested under monotonic loading, and the specimens had three different thickness flanges (7.9 mm, 9.5 mm and 12.7 mm). During the tests, the researchers recorded the applied load, the vertical displacement, and the reaction force on the bolts. The results of the experimental testing showed that 2 failure modes were observed, see Figure 22 and: (1) the bolt shank shear at the contact area with the T- stub flange; and (2) failure by the combination of the bolt shank shear at the contact with the flange and bolt tension failure in the thread.

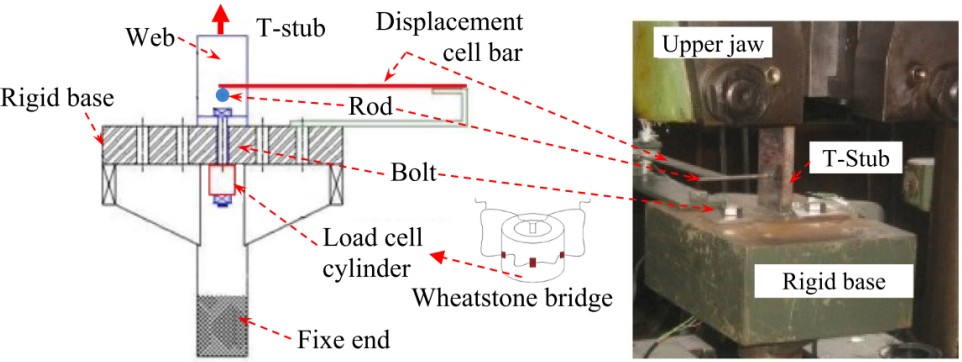

**Figure 21.** T-stub connection attached to a rigid base [47]. "Reproduced with permission from Xulin Zhu, Peijun Wang, Mei Liu, Wulan Tuoya Shuqing Hu, Journal of Constructional Steel Research; published by Elsevier, 2020".

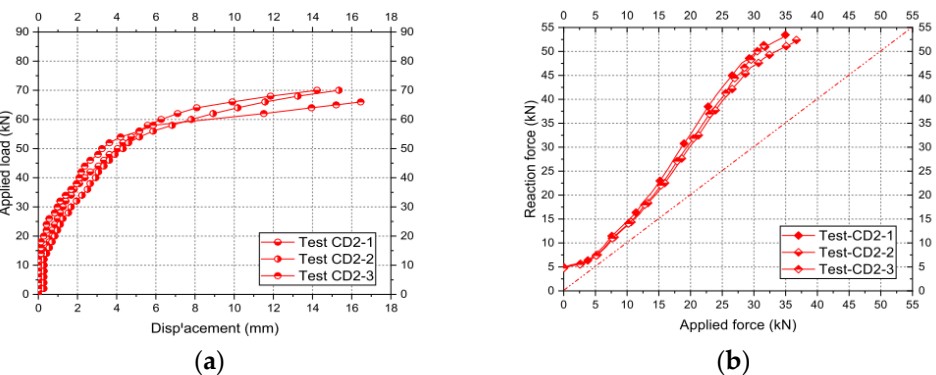

**Figure 22.** (**a**) Force-displacement curve of T-stub specimens of thickness flange equal to 9.5 mm; (**b**) Reaction-force vs. applied load curves of the specimens of thickness flange of 9.5 mm [47]. "Reproduced with permission from Luciano M. Bezerra, Jorge Bonilla, Wellington A. Silva, William T. Matias, Engineering Structures; published by Elsevier, 2020".

In 2020, Qin et al. [48] studied a new type of self-centering connection which solved the common brittle damage problem of conventional steel connections under strong earthquake

conditions (eg. Kobe and Northridge [49,50]). The main characteristic of the new connection is the use of friction T-stubs as energy dissipation devices and the post-tensioning high strength strands, which offered the self-centering capability. The study of this new connection was carried out by the experimental testing of five specimens that were tested under cyclic loads to understand the seismic behaviour of the connection. After the tests were performed, it was observed that the friction T-stubs provided stable energy dissipation ability, and the strands offer self-centering and eliminate the residual drift following a strong earthquake. Furthermore, the investigation proposed an analytical model to determine the yield and ultimate loads.

In 2020 as well, Zhu and Wu [51] published a study about T-stub connections with inserted plates, which allow the T-stub flange to yield under tensile or compression loads (see Figure 23). i.e., these plates enhance the dissipation of energy. The experimental part of this research work tested thirty T-stub connection at the Structures Laboratory at the Fujian Academy of Building Research, which was divided into the following three types of test: nine specimens were tested under monotonic tensile load, ten specimens under compression load and eleven specimens under cyclic load. The experiments showed that two inserted plates did not have a prominent effect on the tensile properties of the T-stub connection. The compressive properties grew with the increase of the inserted plates width. The T-stub under cyclic loads improved its energy dissipation capacity by 50% after the inserted plates are assembled.

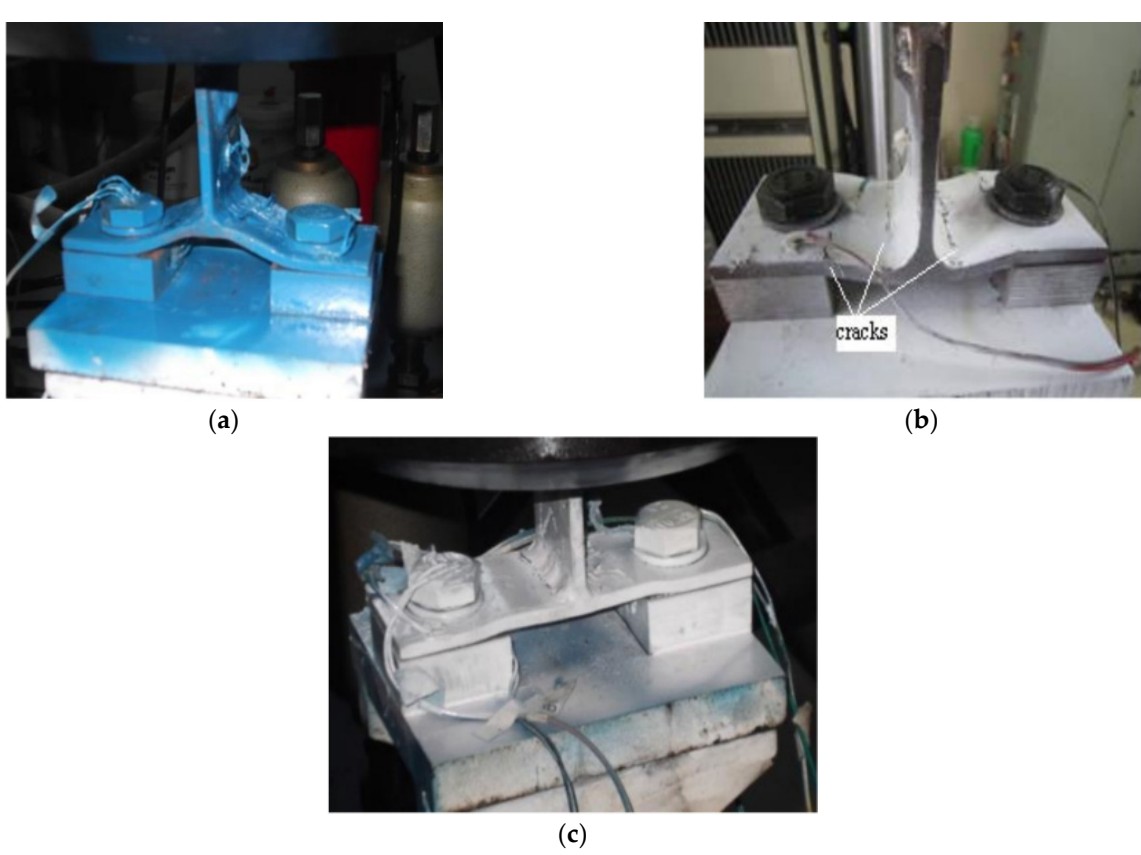

(a)　　　　　　　　　　　　　　　(b)

(c)

**Figure 23.** (**a**) Unstiffened T-stub bolted connection; (**b**) stiffened T-stub bolted connection; (**c**) different orientation of the single T-stub [51].

In 2020, You et al. [52] carried out 25 tests on T-stubs (different flange thickness and bolt diameter) connected by thread-fixed one-side bolts to investigate the tensile behaviour under ambient and high temperatures (500 °C and 700 °C) and steady-state and transient-state (material properties were obtained at steady-state) conditions. In steady-state, the T-stubs were heated first, and when the temperature reached the desired value a load

was applied until the force declined to 80% of the ultimate force. On the other hand, in transient-state, the preselected load (a ratio of 0.5 or 0.7 of the ultimate load) was applied first and kept for 2 min. Then, the T-stubs were heated. The tests were stopped when the T-stub carrying capacity dropped to 95% of the applied load, and this state was defined as the failure limit state. The tests showed that different four failures modes were observed at ambient temperature and elevated temperature. After the experimental tests, in steady-state, an initial stiffness decrement and a ductility increment were identified, with the elevation of the temperature. In transient-state tests, the temperature failure decreased with the increment of the load ratio (from 0.5 to 0.7). Besides, the heating procedure degraded the material of the shape and the bolt differently. Thus, different failure modes were observed at different temperatures (ambient and elevated temperature), e.g., four of the five potential failure modes were recognised. The suitability of use TOB T-stub was analysed by comparing the behaviour of the same T-stub but connected with standard bolts, and the results demonstrated that the ductility and tension strength were nearly similar. Finally, the modified design equations predict the behaviour of the TOB T-stubs at ambient and elevated temperature.

In 2020, Wang et al. [53] studied the behaviour of T-stubs with backing plates at ambient and elevated temperature. Therefore, 30 tests on T-stub with backing plates (different flange thickness) connected by thread-fixed one-side bolts were performed to investigate the tensile behaviour at ambient and high temperatures (500 °C and 700 °C) under steady-state and transient-state (material properties were obtained at steady-state) conditions. In steady-state, the T-stub were heated first, and when the temperature reached the desired value the load was applied until the force declined to 80% of the ultimate force. On the other hand, in transient-state, the preselected load (a ratio of 0.5 or 0.7 of the ultimate load) was applied first and maintained 2 min. Then, the T-stubs were heated. The tests were stopped when the T-stub carrying capacity dropped to 95% of the applied load, and this state was defined as the failure limit state. At ambient temperatures, the tests showed that the backing plate could effectively increase the yield strength and the ultimate strength of bolted T-stubs but the ductility is not affected. Also, two failure modes were observed during the tests, which were completely flange yield and flange yielding accompanied with threads failure. At steady-state, it was observed that the backing plates still offered a strengthening manner for TOB T-subs at high temperatures (due to the fact the hole threads could still be working) even when the yield strength and the ultimate strength decreased at elevated temperatures. Also, the bending deformation of the T-stub flange increased and lateral deformation of bolts at a high level of temperatures. The two failure modes changed at high temperatures, displaying complete flange yielding and flange yielding accompanied by bolt failure. In transient-state, it was observed that the ductility of the TOB T-stub increased when was compared with the same configuration but without backing plates at the same load ratio level. When the load ratio increased to 0.75, it was observed that the failure temperature and failure displacement decreased, which indicated the reduction of the ductility. Additionally, the failure of the hole threads was not observed, and this demonstrated the reliability of the TOB T-stub. Finally, the use of backing plates in TOB T-stubs improved its tension behaviour at ambient temperature, steady-state, and transient-state.

In 2020, Tartaglia et al. [12] studied the T-stub behaviour with preloaded bolts under large deformations due to the significant influence of the bolt preloading on the stiffness, strength, and ductility of the T-stub. Besides, the type of high resistance bolt may affect the behaviour. For instance, the British HR bolt is characterised by the shank necking of the bolt and the German HV bolt is characterised by the stripping of the nut out of the shank under pure tension load. Furthermore, the behaviour of these two types of bolts was studied by D'Aniello et al. [54]. Tartaglia et al. [12] performed 16 tests on T-stubs composed of two sets of eight different geometrical features, which were connected with HR and HV bolts to a rigid support in order to highlight the effects of membrane action in the flange and shear forces and bending moments in the bolts. This membrane action

influences the ultimate behaviour at the collapse of the connection. Indeed, the membrane action transfers large shear forces to the bolt, which cause the ovalisation of the bolt holes and the damage pattern after the test. Additionally, the experimental tests showed that for T-stub that were designed to behave according to failure Mode 1 and 2, the type of high strength bolt HR and HV were not relevant due to the similarities in their behaviour (Force-displacement curve). However, for the T-stubs that failed according to Mode 3, the influences of the bolt failure mode influence the T-stub behaviour.

In 2021, Berrospi Aquino et al. [55] studied the behaviour of three different configurations of T-stubs that were the laminated T-stub (T-W), the fillet welded T-stub (T-FW) and the full penetration welded T-stub (T-FP), see Figure 24. Therefore, two T-stubs of each of the three types (T-W, TFW and TFP) were tested in a universal testing machine. The analyses of the three configurations were performed by employing the force-displacement curve. The results of the test showed that the resistance of T-W is higher than the resistance of the T-FW and the T-FP typology. Additionally, the resistance of the welded typologies could have been reduced for the welding process effect, which typically softens the mechanical properties by around 20%. The typology of T-FP had the highest initial stiffness among the three, and a direct correlation between the initial stiffness and the m factor was observed, see Table 6. Furthermore, the welding process could have affected the microstructure and thus increased the stiffness of the welded configurations. On the other hand, it was observed that the ductility increased while the *m* parameter decreased. Therefore, it seemed that the T-W typology, which has the smaller value of the m parameter, developed the major ductility of the three configurations. The T-FP configuration reached a higher level of ductility between the welded configurations.

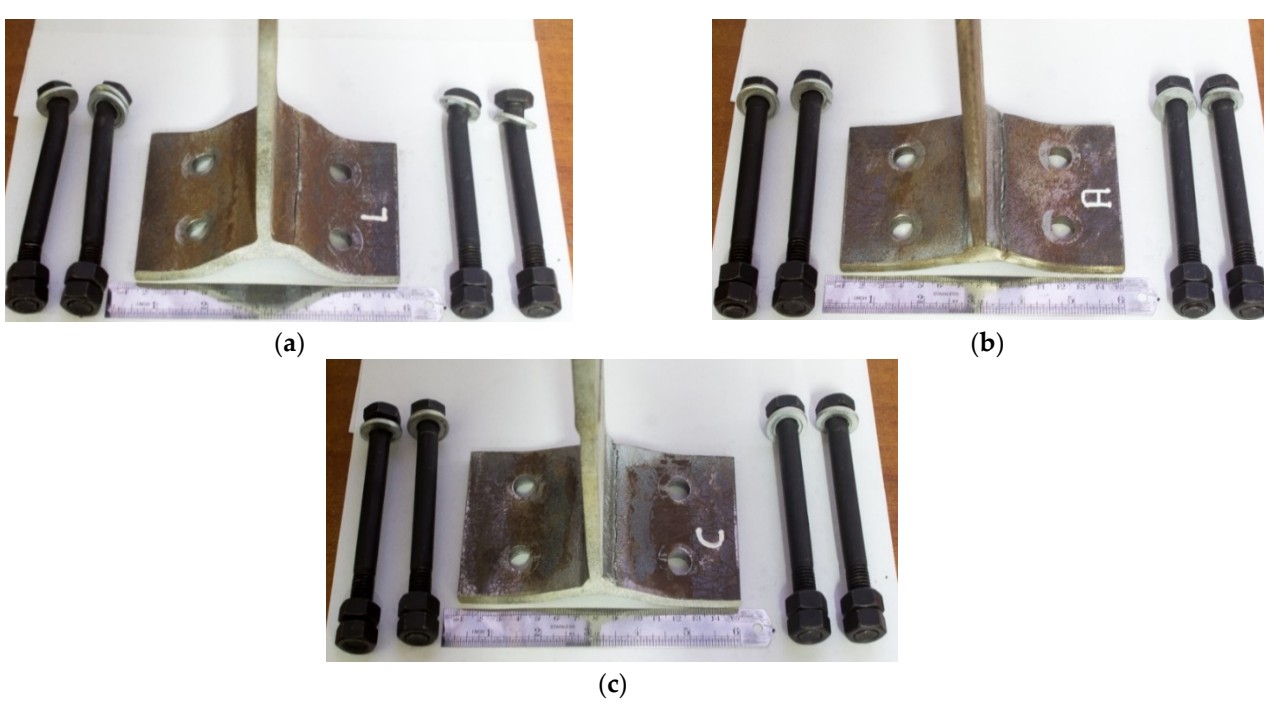

**Figure 24.** T-stubs configurations: (**a**) laminated T-stub (T-W); (**b**) fillet welded T-stub (T-FW); (**c**) full penetration welded T-stub [55]. "Reproduced with permission from Giovani Jesus Berrospi Aquino, Gustavo Alberto Neira Alatrista, Walter Guillermo Loaiza Miranda, et al, ce/papers; published by John Wiley and Sons, 2021".

## 4. Numerical Models

The behaviour of the T-stub component can be predicted with reasonable accuracy by numerical simulations. This numerical simulation requires that the geometry of the joint should be an adequate model, with reliable data to reproduce the constitutive law, the boundary and load conditions should be well assigned and idealised to reproduce the real

behaviour [56]. Since the appearance of the finite element method (FEM) in 1943 [57] FEM is accepted as the most reliable technique for obtaining approximate results for structural mechanical problems [58].

In 1972, the first FEM study of welded beam-column joint was performed by Bose [59]. This investigation covered aspects of plasticity, strain hardening and buckling, and the study results showed good accuracy compared with available experimental data. Since this research publication, many other investigators have used the FEM to study structural joints' behaviour.

In 1976, Krishnamurthy and Graddy [60] simulated the most typical moment connection, the end-plate connection. Also, they were the first to use 8-nodes brick elements to model 3D-joints such as the whole end-plate connection. Although many simulations employ brick elements, not all the simulations were performed with brick elements due to the computational hardware limitations of the 70s. Therefore, the study was focused on founding a correlation factor between 3D elements and 2D elements. Furthermore, in this research work, the authors chose to simulate the whole end-plate connection and not the "tee-hanger" analogy, which is similar to the T-stub component, because according to the AISC, it was not directly applicable to the end-plate connection.

As readers will notice, the studies carried out in 1972, and 1976 are related to T-stub simulations, but not at all because the simulations were done in the complete beam to column connection. Therefore, the following information will be about T-stub simulations. Many of the first research works on this topic were part of the Numerical Simulation working group of the European research project COST-C1, which has the task to do a benchmark for FE modelling of bolted steel connection. Jaspart (some material data of Jaspart test were missed) and Bursi afterwards provided the experimental data for calibrating the T-stub simulations that were labelled as T1 and T2.

In 1995, Bursi and Jaspart [61] studied employing FEM two T-stubs connections (T1 and T2, see Figure 25) that were modeled using an eight-nodes brick element. These simulations were performed in the software LAGAMINE and ABAQUS, and just a $\frac{1}{4}$ of T-stub were modeled for each simulation. The results that they obtained with the LAGAMINE software were used as a point of comparison to calibrate the ABAQUS simulation because the COST-C1 was focused on semi-rigid design procedures.

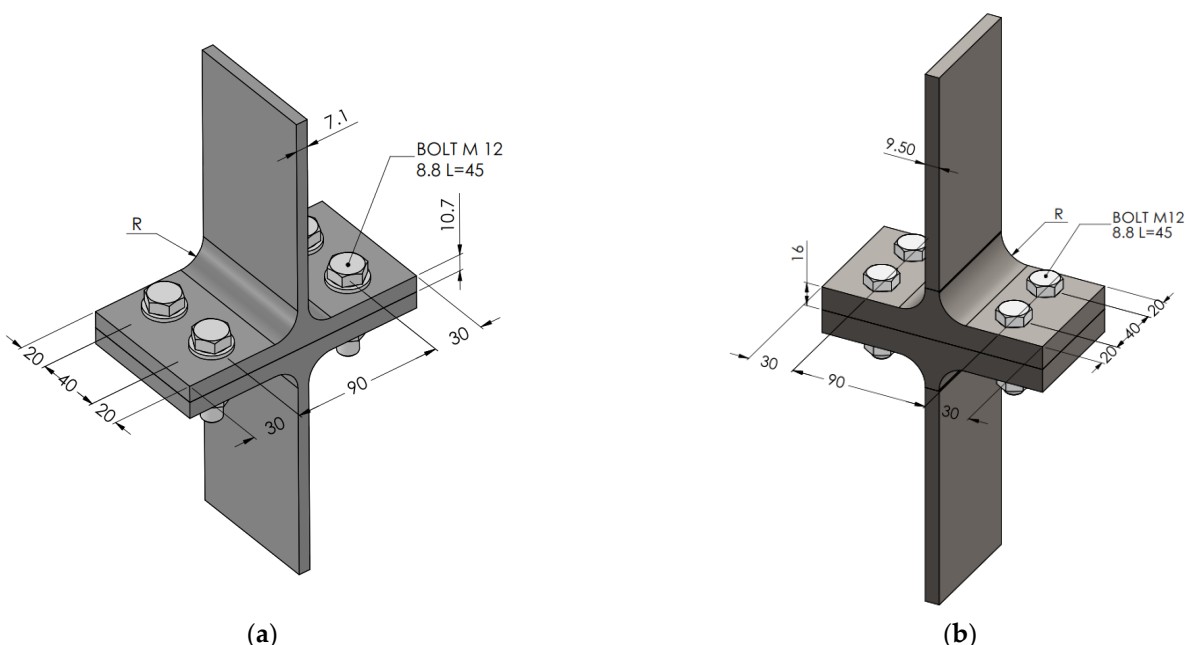

(**a**)　　　　　　　　　　　　　　　　　　　　　　　(**b**)

**Figure 25.** Bursi and Jaspart T-stub geometries: (**a**) T-stub T1; (**b**) T-stub T2. "Reproduced with permission from O.S. Bursi, J.P. Jaspart, Journal of Constructional Steel Research; published by Elsevier, 1997".

In 1997, Jaspart and Bursi [4] continued investigating the same two T-stub connections (T1 and T2). This work was focused on presenting a study of the plastic behaviour of elementary T-stubs and proposing these simulations as a benchmark for the validation of FE software packages. Furthermore, in the same year, they also published an investigation [62] about the calibration of the T-stub T1. The calibration was performed in ABAQUS code on test data results, and the parameters that were considered are: the influence of the brick element type (C3D8, C3D8R, C3D8I), the influence of friction coefficient between the flange and the foundation (mu = 0, mu = 0.25 and mu = 0.5) and the bolt model influence (3D model and spin model). As a result of the simulations, they concluded the following:

- The C3D8I element reproduces the behaviour of the T-stub connection more precisely.
- The friction coefficient affects T-stub responses only on the large displacement regime.
- The spin bolt model with two beam elements gives more accurate results than the model with only one beam element when the models are compared with the results of the 3D bolt model.

The same year, Mistakidis [63] proposed a numerical FE 2D model capable of describing plasticity, large displacement and unilateral contact effects. The simulation results (force-displacement curve) showed that they had not achieved a good agreement with the experimental curve. However, the simulations were improved by changing the constitutive law of the material because no data for the ultimate stress value was available.

In 2002, Swanson studied the behaviour of T-stub connections [64] employing a robust FE model. These simulations were intended to supplement the experimental part of the research work and provide insight into the T-stub behaviour and the stress distribution. The simulation results were used as a point of comparison to validate a 2D finite element model. Moreover, the authors analysed the bolt responses (different pretension loads) and the prying effect. Even though the simulations reached a good agreement with experimental results, the material law of the T-stub had been considered with nominal values because the material characterisation had not been done until the FE model was finished. Therefore, this procedure is considered questionable.

In 2003, Gantes and Lemonis [65] performed an analysis of the influence of the equivalent bolt length in finite element modelling of T-stub steel connections. The simulations were performed in the software MSC/Nastran and calibrated with experimental data obtained from the experimental tests of the T-stub labelled as T1 and T2 of the Jaspart and Bursi investigation [4]. Because of a significant lag in the maximum displacement observed in the calibrated simulations, a parametric analysis was performed by changing the bolt length, which was initially determined by Aggerkov's expression. The parametric study showed that the required correction in the bolt length is heavily dependent on both the applied preload level and the developed failure mechanism. For instance, for the non-preloaded T-stub T1, the experimental displacement was reached when the bolt length increased by 50% or for the preloaded T1, the experimental displacement was reached when the bolt length increased by two times.

In 2004, Girão et al. [66] did a finite element model to study the non-linear behaviour of bolted T-stub connections, which is the component that idealises the tension zone of the end-plate connection. The simulations were performed in two different T-stub connections: rolled profiles cut along the web and two plates, flange and web, welded to each other in a T shape with a continuous fillet weld. The experimental test results, which were aforementioned in the experimental testing section, were used to calibrate the simulations. The results of these simulations unveiled that the FEM of the rolled T-stub, see Figure 26a,b, reproduced accurately the behaviour of the connection. However, the fillet-welded T-stub simulations showed that the difference between numerical and experimental results was considerable due to the residual stresses and the modified material properties close to de welded toe (HAZ). Therefore, to reproduce the T-stub behaviour more accurate, the material properties were softening around 20% in the HAZ zone in the numerical simulations, see Figure 26d. This research was also focused on the prediction of more T-stub geometries. Therefore, after the calibration, a parametric studied was performed, see Figure 26c.

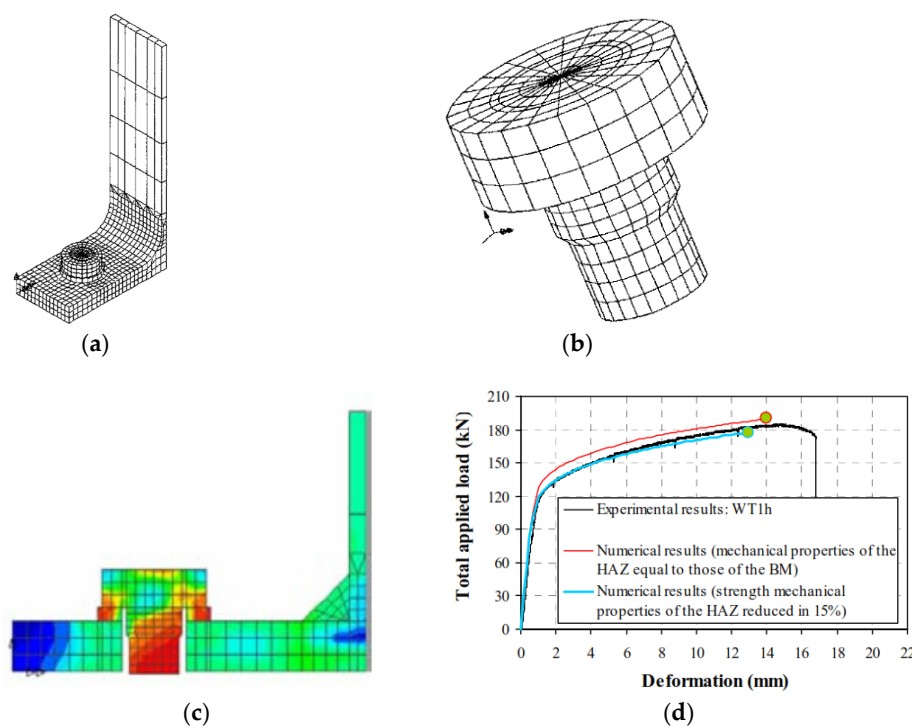

**Figure 26.** (**a**) FEM of the rolled T-stub; (**b**) bolt discretization; (**c**) FEM of the parametric study of the fillet welded T-stub; (**d**) Force-displacement curve of the fillet welded T-stub [66].

In 2007, Al-Khatab [67] investigated the effects of backing plates (BP) used to reinforce beam to column connections. The T-stub, which is the main component in bolted joint in bending, was analysed in the non-reinforcement stage to calibrate the FEM, see Figure 27a,b. This calibration was done in two types of FE models: (1) a 2D model (2) a 3D model, which were both compared to define the limits of the 2D model. After the calibration, a parametric study was conducted to examine the prying forces' relevance and the evolution of the contact areas under the T-stub. Afterwards, the study was focused on investigating the effects of the BP thickness and the bolts preloaded. On the one hand, the results of the 2D simulations showed that this type of model is suitable to analyse the T-stub without backing plates, see Figure 27c, and gave satisfactory results for short T-stubs, see Figure 27d,e, with backing plate. However, the resistance is a little underestimated. On the other hand, a long T-stub, see Figure 27f, is more suitable for being analysed by 3D FEM. They concluded that the T-stub resistance and the initial stiffness increase with the thickness of the backing plate, according to 3D models. EC3 prediction of the T-stubs with backing-plates gives satisfactory results for the resistance compared to numerical results.

In 2010, Wang et al. [15] investigated T-stub connections with blind bolt Hollo bolts. This research investigated the effects of this type of bolts on the initial stiffness, strength and ductility. Thus, two 3D numerical models were developed, see Figure 28. The first was the T-stub connection model with Hollo bolt, which was reckoned for studying the clamping effect, the force transfer mechanisms, stiffness, strength and deformation capacity of the connection, and the other was developed for studying the standard bolt. The models were calibrated with experimental data (Hollo bolt T-stub model [68]) before the results of both models were compared to understand the behaviour changes. Afterwards, a parametric study was performed, and the parameters that were taken into account are the following: angle of the flaring sleeve ($\beta$), bolt shank diameter ($d_{sh}$) and flange thickness ($t_f$). The main conclusions of the investigation are that the T-stub connections made with the blind-bolt results in less initial stiffness than those with standard bolts. According to the parametric study results, the flange thickness most influences the strength and stiffness of the blind-bolted connection compared to the flaring sleeve angle and the bolt shank diameter.

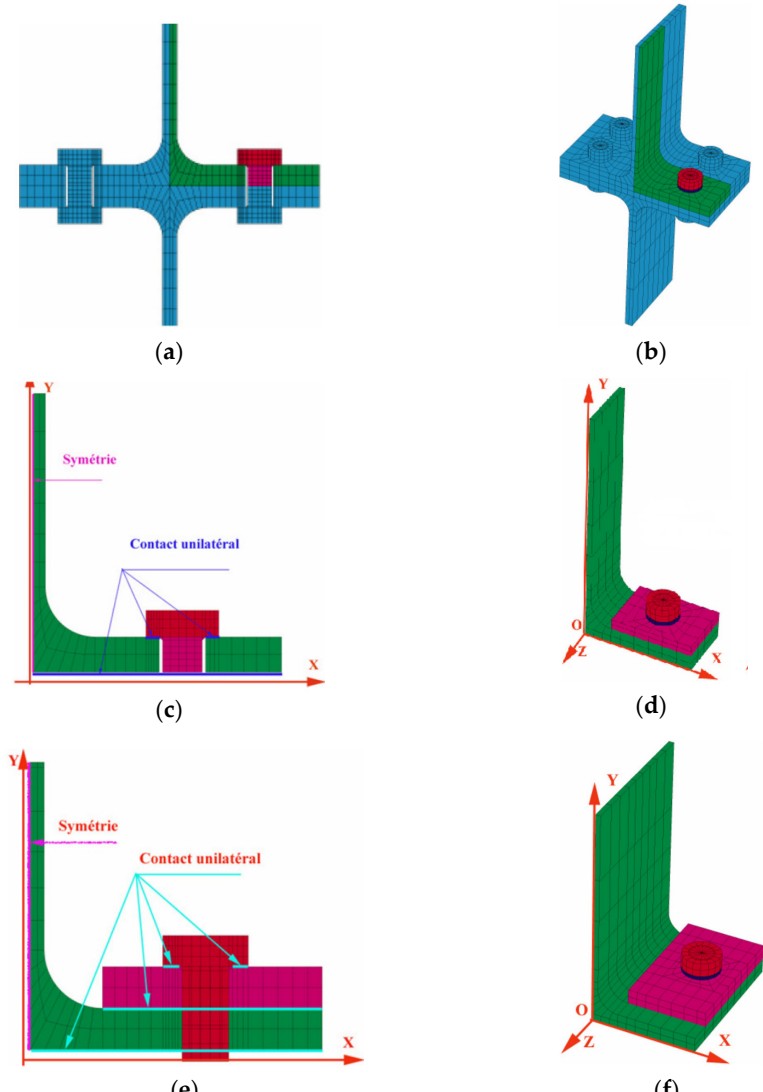

**Figure 27.** Finite element models of the T-stub: (**a**) Non-reinforcement 2D T-stub; (**b**) Non-reinforcement 3D T-stub; (**c**) Quarter of non-reinforcement 2D T-stub; (**d**) Quarter of Reinforcement 3D short T-stub; (**e**) Quarter of Reinforcement 2D short T-stub; (**f**) Quarter of Reinforcement 3D long T-stub [67]. "Reproduced with permission from Z. Al-Khatab, A. Bouchaïr, Journal of Constructional Steel Research; published by Elsevier, 2007".

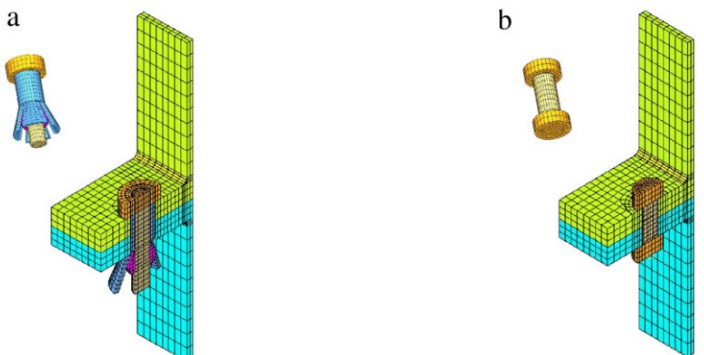

**Figure 28.** 3D finite element mesh: (**a**) Blind-bolted T-stub; (**b**) Standard bolted T-stub [15]. "Reproduced with permission from Z.Y. Wang, W. Tizani, Q.Y. Wang, Engineering Structures; published by Elsevier, 2010".

In 2010, Loureiro et al. [69] developed two different FEM models to determine important variables of the T-stub, which were used to propose an analytical model that took into account compatibility requirements and prying forces. The first model was calibrated with the experimental result of Faella (TS11) and was discretised using eight-nodes brick elements. This model was used to characterise the deformation response and the distance between the plastic hinges in the T-stub flange. Thus, the results showed that the *m* parameter was quite similar to the EC3. The second was a FEM plate model of half T-stub geometry, which was validated with the experimental results of Faella (TS22, TS11). After the calibration of the model, a parametric study was performed to determine the effective length.

In 2011, Tanlak et al. [70] started an investigation with the objective of developing computationally efficient and accurate finite element models for bolted joints under impact loading. Therefore, a 3D detailed FEM model of the T-stubs was developed to analyse the T-stub behaviour without considering the computational cost. The element type C3D8R of the detailed model was chosen due to the large strain and large deflection capabilities. Additionally, the element allowed the reduced integration for computational effectiveness. The reduced integration does not create stiffness in some deformation modes and the hour glassing effect may occur. Afterwards, simplified models were developed to reduce the computational cost. Therefore, (i) the full model with shell plate were analysed (the same considerations for the bolt, but the flange was discretized using shell elements), (ii) rigid shank with coupling constrains (shell elements for the flange, the bolt shaft was considered rigid and coupling constrains were assigned to simulate the effect of bolt head and nut), (iii) deformable shell bolt (shell elements for modelling the nut and bolt assembly), (iv) rigid shell bolt (the same of iii, but the nut-bolt assembly is modelled with rigid shell), (v) Timoshenko beam with coupling constraints (the bolt shaft is modelled with a Timoshenko beam 3D node quadratic element), (vi) Timoshenko beam with coupling constraints without a hole (the flange did not have bolt hole, the other elements are the same of v and the coupling constrain was assigned), (vii) tie constraint with hole (the force transfer between the frame and the plate was achieved through a tie constraint defined between the inner surfaces of the sheets in the region compressed by the washers), (viii) tie constraint without a hole (the tie constraint was used to model the clamping effect of the bolt-nut assembly), (ix) cross-coupling constraint (a kinematic coupling constraint is used to simulate the force transmission between the sheets), (x) connector beams along the perimeter of the hole (beam type elements were defined in the perimeter of the hole), (xi) connector beams along the perimeter of the hole and the washer's outer profile (similar to x, but beam elements were assigned to the edge of the washer's profile), (xii) cross connector beams (12 connector beam-type elements that joined the midpoint of the line between the centres of the holes and nodes at the perimeters). Finally, all the models were tested with the same mesh density and that showed that only the simplified model (i) did not save computational time. Moreover, the simplified model (iii) was the most accurate to predict the T-stub behaviour for different loading cases and mesh densities.

In 2014, Abidelah, Bouchaïr and Kerdal [71] investigated the relevance of influence of the bolt bending on the behaviour of T-stubs. Therefore, a 3D numerical model was developed in the software Cast3m and validated with experimental results found in the literature. These FE models were used to predict the behaviour of T-stub connections, quantify the axial and bending loads in the bolts. The numerical models took into account the non-linearity due to the plastic behaviour of the material and the evolution of the contact area between the two flanges. A parametric study was performed to evaluate the influence of the bolt bending moment on the behaviour of the T-stub by varying the flange thickness and the bolt diameter. Afterwards, an analytical model was proposed, which considered the bending of the bolt and the axial force. The main conclusion of this study was the following: The FE analysis of the T-stub showed that the bending in the bolt starts since the beginning of the load. When the bending load is taken into account, the ultimate resistance of the bolt is reached. In the contrary case, the bolts reach about 70% of their

ultimate resistance. The increase in T-stub flange thickness leads to the decrease in the value of the bending moment in the bolts that will, therefore, develop their ultimate resistance.

In 2015, Francavilla et al. [72], because of the lack of information to predict the ductility of T-stubs connection in the codes such as EC3 [1], developed a simplified 2D FEM model of a bolted T-stub, with only one bolt row, in the widespread commercial software SAP2000 to estimate the plastic deformation capacity. This model was made by using beam elements. The calibration of the FEM model was done with experimental data of specimens tested at the Material and Structure Laboratory of Salerno University in 2001. The authors concluded that the 2D FEM model is accurate enough to predict the stiffness and resistance and the plastic deformation capacity. Furthermore, it allows of been developed in commercial software such as SAP2000.

In 2016, Cabaleiro et al. [14] studied the behaviour of T-stubs connected with clamps by changing the clamp geometry. This type of connection has the advantage of been dismountable and reconfigurable. Nevertheless, the information that could be found in the literature about it was insufficient. Therefore, a FEM model was established to validate an analytical model that could predict the behaviour of clamp joints. This new method is based on the EC3 method. The calibration of the FEM model was performed against experimental tests results (T-stubs obtained from IPE220 profiles), and the most highlighted of the simulation was the used of tetrahedral finite elements (see Figure 29b) for meshing the T-stub and the clamps. The results of the investigation (IPE220 experimental, FEM analytical and HEA200 analytical and FEM) showed that the proposed method was suitable for analysis this type of connection, depending of the geometry parameters of the clamps (lever arms). According to this, a greater front lever "a" (see Figure 29a) of the clamp improves the T-stub behaviour and impair the bolt behaviour. Besides, the increment of the rear lever "b" (see Figure 29a) reduce the axial force in the bolt. Moreover, the comparison of the analytical results and the FEM simulations showed that the analytical model values were 10% lower than those found in the FEM.

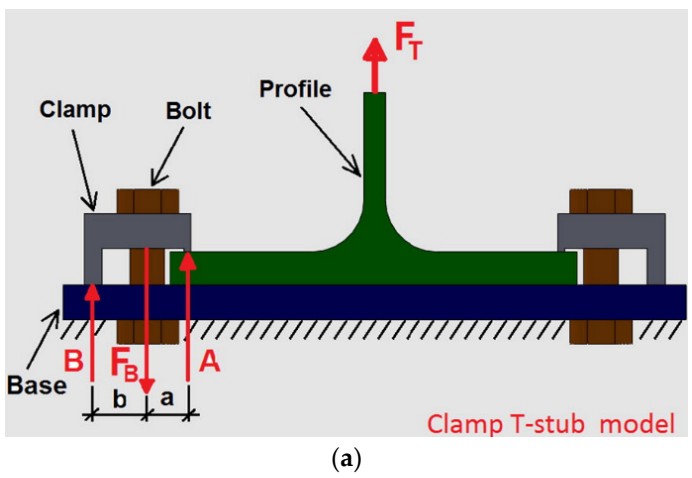
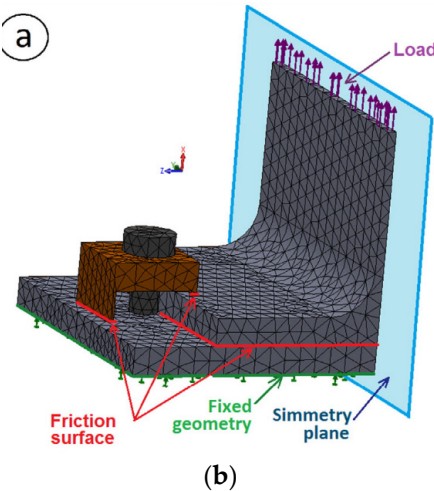

(**a**)    (**b**)

**Figure 29.** Clamp T-stub: (**a**) Geometric parameters; (**b**) Tetrahedral mesh [14]. "Reproduced with permission from Manuel Cabaleiro,Belén Riveiro,Borja Conde,José C. Caamaño, Journal of Constructional Steel Research; published by Elsevier, 2016".

In 2016, Ribeiro et al. [73] studied the response of T-stub connections under impact loads. This assessment was developed by means of the finite element method. In the FEM model, the T-stub assembly was simplified by employing symmetry conditions, and it was composed of a rigid base, the T-stub (flange), the bolt (the head, shank, nut and washer), and the pulled-out plate (web). The constitutive law of the material is enhancement due to the impact load condition. This increment of the constitutive law is obtained by adopting the dynamite increase factor (DIF), which basically gives the relation of the dynamic

strength to the strength obtained under static conditions. The principle of Hooputra [74], which is included in the ABAQUS software package, predicts the failure of the FEM model. This model assumed that two main relevant mechanisms induced fracture of a ductile metal: ductile fracture due to nucleation, growth and coalescence voids and the shear band localisation cause the shear fracture. The authors considered only the ductile fracture for the investigation, and the von Mises criterion was adopted. The validation of the FEM models was done against quasi-static experiments performed by Barata et al. [27] and Ribeiro [19], and the dynamic calibration was performed by comparing the experimental result of the first impact of 120 bar of T-10-D120-160 and the impact of 160 bar of the T-10-D160 test (the test of this investigation). Afterwards, a parametric study was conducted to study the influence of the maximum applied load, the impact duration, and the T-stub flange thickness. The investigation results showed that the dynamic load level value did not significantly affect the force-displacement curve. However, the load application time has a larger effect on the response. Finally, the increment of the T-stub flange result causes the increment in the stiffness of the T-stub, but the decrement of the force-displacement enhancement, more precisely, for T-stubs having failure Mode 3.

In 2017, Šliseris et al. [75] performed a numerical analysis and experimental validation of a type of T-stub steel joint with preloaded bolts. The numerical analysis took into account large strains, non-linear plasticity and contact mechanics. The calibration of the connection was conducted at the Riga Technical University. The parametric study took into account the flange thickness (20, 30, 40 mm) and the preloading of the bolt (0%, 25%, 50% and 70%). The investigators concluded that the level of the bolt preloading and the flange thickness influence the failure modes. By increasing the thickness of flanges up to 40 mm, nearly equal stress distribution in all bolts was obtained. The joint with 70% preloaded bolts has around a 30% higher ultimate load than the joint with 0% preloaded bolts.

In 2018, Wulan et al. [17] analysed the failure mechanism of thread-fixed one-side bolted T-stubs under tension by a finite element model to provide basic knowledge for its potential application in bolted beam-column endplate connection. The FEM model was calibrated with the experimental results carried out by Liu et al. [16]. After the models were calibrated with the tests T16-14 and T27-140, a parametric study was performed. It took into account the effects of the steel grade, the bolt diameter, and the bolt spacing on the T-stub behaviour. The main conclusions of the investigation were the following:

- The simulation was in good agreement with test results, which meant that the proposed FEM could be applied to investigate the thread-fixed one-side bolted joints' behaviour.
- The proposed equations were in good agreement with the FEM simulation. However, the yield strength calculated from the load-displacement curve was slightly higher than that predicted by the proposed design equations, which means that the proposed design methods predictions are on the safe side.
- The parameters that were taken into account in the parametric study (bolt diameter, bolt spacing, and flange yield strength) affected the behaviour of the connection (the yield strength and the failure mode). The FEM model and the design equations proved that an undesirable presence of thread shear failure (Mode 3b) and the Mode 2b did not appear.

In 2019, Gödrich et al. [76] were focused on a design approach of T-stub component by using the component-based finite element method (CBFEM), which combines the analytical component method and the finite element method (FEM) (distribution of internal forces). The results of the CBFEM model were verified with the component method. However, when the comparison between both models had not reached a good agreement, the FEM model was used to verify the results. This FEM model is called research oriented model (ROM), which was calibrated with experimental data collected by the authors and data found in the literature. After the validations of the CBFEM models were done, the researchers concluded that this model reproduced with reasonable accuracy the T-stub behaviour with an error of no more than 10%.

In 2019, Bao et al. [46], as was mentioned above, studied the mechanical behaviour of bolts used to connect T-stubs. For this, six tests were conducted and then one of the test results data was utilised to verify a 3D FEM model. The calibration results showed that the model reproduced with good accuracy the behaviour in terms of deformation, axial tensile stress, bending moment. Afterwards, a parametric study of 17 simulations was performed to evaluate the influences of the T-stub flange thickness, inner flange length, outer flange length and bolt diameter. The results of the simulations showed that when the inner length of the flange increased, it improved the prying force and the bending moment, and the increment of the flange thickness decreased the prying force and the bending moment. Besides, increasing the bolt diameter provokes the increment of the prying force. Finally, the parametric study results were used to verify a new T-stub connector model and a calculation method, which not require the calculation of the prying force.

In 2020, Bezerra et al. [47] studied the behaviour of the aforementioned T-stub connection connected to a rigid base. The calibration of the FE models was performed in the ABAQUS software, and the experimental results were used for this purpose. After the calibration had reached a good agreement with the experimental results, a parametric study was conducted to observe the effect of the flange thickness on (a) the contact stress distribution between the flange and rigid base, (b) the prying actions on bolts, and (c) the shear stresses on bolts. The simulations results showed that the behaviour of the connections was reproduced with good accuracy (Force-displacement and applied load-reaction force in the bolts), and the results in the parametric study concluded that the prying action is higher for flange thickness of 7.8, 9.5 and 12.7 mm. However, for lower values, the predominance of the shear stress on the bolts is notorious.

In 2021, Berrospi Aquino et al. [55] developed three FEM models to study the influences of three types of connection between of the web and flange in T-stub connections. The three types of connections were the laminated T-stub (T-W), the fillet welded T-stub (T-FW) and the full penetration welded T-stub (T-FP). The FEM models were calibrated against the experimental data of the experimental part of the study in order to reproduced the T-stub behaviour. The results of the investigation showed that the higher stresses were found near the interface of the web and flange. Additionally, it was observed that the EC3 underestimated the resistance of the T-stub.

In 2021, Jiménez de Cisneros et al. [77] studied the T-stub component by developed a numerical model that was analysed with the meshless method. The meshless method is based on the external approximation (energy functions) that considered each part of the elements of an assembly as a finite element. This energy functions are infinite outside the boundary of the Sobolev space and they have a finite energy in the boundary of the Sobolev space (convergence). The meshless software that was employed in this investigation allowed to analyse the assembly without simplifications (which is usually done for FEM models), and also it saves the time that is spent in the mesh convergence due to it is not require. The results of the investigation (which were compared against experimental data of [4]) showed that the meshless model can reproduced the T-stub behaviour with good accuracy in the elastic zone. However, the prediction of the T-stub behaviour in the plastic zone was not good enough to consider the meshless software suitable for this kind of analysis.

## 5. Informational Model

Different methods of studying the T-stub were presented (analytical, experimental testing and numerical models). With the recent advances in statistical, machine learning and artificial intelligence techniques, more sophisticated approximation models have been developed in engineering domain investigations [78].

Informational models could be developed based on different methods such as fuzzy logic methods, neural networks (NN), artificial NN (ANN), evolutionary algorithms, support vector regression (SVR), probabilistic reasoning, Bayesian methods and statistical learning procedures, etc. These methods are covered by soft computing (SC) techniques,

which is an emerging and more or less established family of problem-stating and problem-solving methods that aims to mimic natural intelligence [79].

The artificial neural network (ANN) is among the most widely known approximation models. Various studies were done about the beam to column connection. For instance, in 1996, Jadid and D.R. Fairbairn [80] estimated the response of a cast-in-situ beam to column joint (a beam to column joint that is made of concrete) based on 34 tests. This research aimed to demonstrate a concept and a methodology of parallel distributed processing-based learning in artificial neural networks.

In 1997, another group of researchers, Stavroulakis et al. [81], proposed a two-stage neural network approach for the elastoplastic analysis of steel structures with semi-rigid connections. In the first stage, the moment-rotation law is obtained as a quadratic programming problem (QPP) by the first neural network, based on the perceptron model, from the six experimental test results. In the second stage, the second NN, based on the Hopfield model, resolved the resulting QPP. This research work demonstrated that ANN models are also able to accurately estimate the response of single web-angle bolted con- nections.

In 1997, Anderson et al. [82] used ANN to predict the moment-rotation curve of the minor axis of end-plate connection between the beam and column. The results of 20 tests provided the training of the ANN, and the variables that were taken into account were the column depth of section, flange thickness and web thickness. The beam flange breadth, depth of the section. The connection number of bolts and plate thickness.

In 2005, De Lima et al. [83] studied the behaviour of the welded end-plate and double angle connections by ANN. The author addressed that the initial stiffness and the bending moment resistance were accurately estimated with only two ANNs for each connection type.

Most of the research studied the whole connection, such as the end-plate connection, since this method appeared. However, in 2014, Ceniceros et al. [78] studied the T-stub component by a numerical-informational method and predicted the response of the component. This hybrid approach constitutes a fusion between the FE method and metamodels based on soft computing (SC). The numerical informational method's methodology starts when a representative dataset (geometrical parameters and mechanical properties) of T-stub configurations are generated using design of computer experiments (DoCE). Then this representative dataset is used as inputs in a refined FE model to run a parametric simulation. Afterwards, the simulations provide the force-displacement curves of the dataset and are stored then. These curves are characterised into a set of physical meaning parameters, and this information is used in the following learning process. The last process consists of training and testing metamodels by using a combination of SC techniques based on SVR and genetic algorithms (GA), which were employed to achieve overall and parsimonious metamodels. Finally, the parsimonious metamodels were tested against new data to assess both their prediction capacity and generalisation ability, see Figure 30.

## 6. T-Stubs of Non-Conventional Steel and Other Materials

In 2000, De Matteis et al. [84] focused on the study of the behaviour of aluminium T-stubs. Thus, the authors carried simulations that were calibrated with available data from the literature. At that moment, no experimental tests on aluminium T-stubs were carried out; thus, the 3D eight nodes (C3D8R) FEM model was calibrated with experimental data of steel T-stubs. The numerical analyses showed a wider range of failure mechanism for the aluminium T-stubs than steel T-stub. Failure mode 2 was divided into three. Furthermore, the HAZ effect was taken into account because, for some aluminium alloys, the mechanical properties are softening by 50% due to welding. The simulations result that did not consider and considered the HAZ were compared. This comparison showed that the strength of the material was reduced. The analysis also showed that the collapse mechanisms were not different from the steel T-stub.

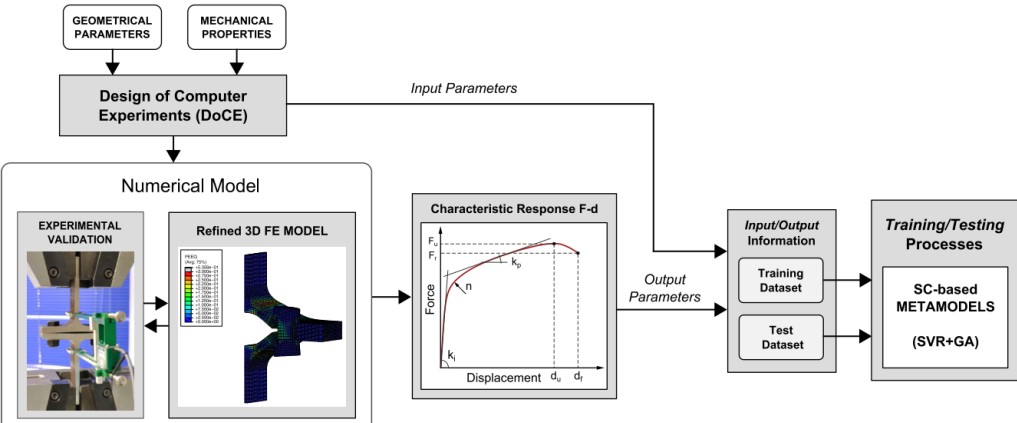

**Figure 30.** Scheme of the proposes of the numerical-informational model [78]. "Reproduced with permission from J. Fernandez-Ceniceros, A. Sanz-Garcia, F. Antoñanzas-Torres, F.J. Martinez-de-Pison, Engineering Structures; published by Elsevier, 2015".

In 2006, Abolmaali et al. [85] studied the hysteresis behaviour of T-stub connections using shape memory alloy (SMA), where the tee and the bolts were SMA, by comparing the results with the behaviour of steel T-stubs. The investigation was performed in two phases: phase I was focused on determining the optimum heat treatment temperature to establish a superelastic effect. Therefore, the material was heat-treated at 300 °C and 350 °C and three different testing protocols were employed: (1) monotonic tensile testing; (2) cyclic testing, and (3) Tensile testing of the cycled parts. Phase II tested cyclic T-stubs connection made of SMA and steel. The optimum temperature determined in phase I was employed to treat the SMA fastener of phase II. The comparison of the experiments showed that the energy dissipation of T-stub with SMA fastener was higher than those obtained with steel material for the particular stress level under consideration. The steel test was stopped when it reached the same stress level that the SMA specimens developed, which failed in early stages of loading at 28th and 22nd strain cycles.

In 2006, De Matteis and Mazzolani [86] studied the behaviour of aluminium weld T-stubs with different geometrical configurations and material properties. The T-stubs were obtained by welding together three different wrought aluminium alloys, which are AW 6061 (t = 10 mm), AW 6082 (t = 12 mm) and AW 7020 (t = 12 mm). 26 T-stubs were obtained by combining the different available materials and were tested under a monotonic load and a cyclic load. Afterwards, the results of the monotonic tests were compared with the analytical k-method (found in EC9) and with the EC3 method. This comparison showed that many factors such as the geometrical, mechanical and factors as the parasitic bolt bending, effects of the HAZ affect the prediction of the behaviour of the analytical method. Despite the things mentioned, the analytical k-method can be considered fairly reliable, conservative and suitable for design purposes due to its simplicity. Moreover, it was determined that the appropriate choice of mechanical parameters the values of strength predicted with the k-method are always on the safe side.

In 2009, De Matteis et al. [87] studied the behaviour of aluminium welded T-stubs under monotonic loads. The investigation was focused on developing a FEM model capable of reproducing the behaviour of the T-stub. Therefore, 26 experimental tests [88,89] with different geometries (flange thickness, bolt number and distribution), three different heat-treated wrought aluminium alloys (AW 6061–AW 6082–AW 7020), and three types of bolts (aluminium and steel bolts) were employed to calibrate the model. This numerical model used hexahedral elements (C3D8R) for the T-stub and tetrahedral elements for the bolts. Additionally, the constitutive law of the heat-affected zone and the unheated-affected zone of materials were introduced as a multi-linear curve in the simulation. The simulation results showed that the FEM model reproduced with good accuracy the tests in terms of stiffness, resistance and ductility. Finally, the researchers evaluated the viability of the

EC9 for predicting the T-stub behaviour. Therefore, the EC9 predictions were compared against experimental data. This comparison showed that the EC9 predictions were slightly conservative in predicting the strength, and the prediction could be improved through a more precise evaluation of the effective width.

In 2012, G. De Matteis et al. [90] developed a FEM model which was calibrated with available experimental data from the literature [86] and performed a parametric study of 43 simulations of welded T-stubs made of aluminium, which had different configurations and types of bolts (grade 4.8, 10.9 and 7075 alloy bolts) to analysis all the possible failure modes. The simulations were carefully developed to compare with the EC9 and to check the reliability of the design standard method. The EC9 proposes four failures modes to describe the behaviour of welded T-stub made of aluminium. The comparison of the parametric study results and the prediction of the EC9 showed that the EC9 k method predicted with good accuracy the behaviour of aluminium-stubs connected with weak bolts, and the EC9 overestimated the ultimate strength when thick T-stub flanges and large bolt pitches are used, especially when failure mode 1 appears because it is quite related to the "effective length" concept. Additionally, the authors proposed a further study to check the reliability of the EC3 formulation for the "effective length".

In 2017, Chen et al. [91] studied the effect of the welding on the tensile performance of high strength steel (HSS) RQT S690 T-stub Joints. Therefore, three types of T-stub (flange thickness equal to 8, 12 and 16 mm) made of RQT S690 and one type of T-stub (flange thickness equal to 16 mm) made of normal strength steel (NSS) S385 were fabricated twice (a total of eight T-stub) according to AWS structural steel welding code. Additionally, to the experimental testing, the investigation was focused on developing two FEM model (for each type) that could predict the behaviour of the T-stubs. The first FEM model did not consider the effect of the welding, and the second model took into account the effect of the welding (HAZ). Therefore, the first step was to perform a transient thermal analysis employing a FEM model to obtain the time-depending temperature distribution field, where the heat source that was utilised for the simulation was the body heat source model base on Goldak's double ellipsoid heat source model theory [92]. Secondly, the time-dependent temperature distribution data was used as a thermal loading in the mechanical analysis to simulate the alteration in the mechanical properties and subsequent tensile test. This investigation determined that the experimental Force-displacement curves of the T-stub of two different materials had the same pattern. Also, the suitability of the EC3 design resistance prediction equation was evaluated, and it was observed that the EC3 accurately predicted the first yield resistance of the T-stub made of S385 but overestimated the behaviour of the RQT S690 T-stub. This overestimation was attributed to the changes in the material's mechanical properties due to the welding process. The FEM analysis confirmed this attribution, and it was observed that the FEM model that considered the effects of welding reproduced with more accuracy the T-stub behaviour than the T-stub that not considered the welding effect.

In 2018, Sun et al. [93] studied the effect of welding and complex loads on T-stubs made of high strength steel (HSS). The main HSS microstructure is composed of martensite and bainite, which are not stable at high temperatures [94]. During the welding process, the material reached high temperatures that affected the microstructure in the heat- affected zone (HAZ). Because of all the aforementioned, the researchers performed six tensile tests on double T-stub and demonstrated that the EC3 [1] formulae (Equation (1), which are accepted for determining the failure modes of conventional steel, cannot be used directly for design HSS T-stubs. Therefore, a FEM model that considered the effect of the HAZ, was employed to predict the behaviour of the T-stub, and the results showed that the alternative (Equation (2)) formula of the EC3 predicts with good accuracy the first yield resistance. Furthermore, the T-stub under combined axial and shear loads were investigated by the FEM. The results showed that the strain concentration is in the tensile side of the T-stub under combined loads, and the compression side of the T-stub did not achieve Mpl,1. Besides, the ultimate resistance and first yield resistance decrease with the decreases in the

beam height. Moreover, the investigation proposed a formula, which considered a factor of the combined axial and shear load, to predict the T-stub behaviour.

In 2019, Carazo et al. [95] studied T-stub components obtained by additive manufacturing (AM) using polymers. These T-stubs were printed by the fused deposition modelling (FDM) technique and made of four different materials: polylactide (PLA), acrylonitrile butadiene styrene (ABS), carbon fibre-reinforced polyamide (PA CF) and carbon fibre- reinforced polyethylene terephthalate (PET CF). This research aimed to study the behaviour of the components and compare the experimental force-displacement curves with the prediction of the Eurocode 3 [1]. As a result of this comparison, they determined that the code prediction was not accurate enough.

In 2019, Wang et al. [96] carried out 30 tests of extruded aluminium T-stubs specimens connected by swage-locking pins under monotonic loads. The different specimens were designed by varying the distance between the pins and the web (m + 0.8r), the flange thickness, the swage-locking pin collar type and the diameter and layout of the fasteners. Before the T-stub tests were performed, 42 experimental tests on the swage-locking pins were carried out under tension, shear, and combined load (shear and tension) to assess their load-carrying capacity. The tests showed that the swage-locking pins developed a new failure mode (compared to standard bolts) that was called collar pull-out. Additionally, the 42 tests were complemented by the tensile coupon tests on the T-stub plate and pin material. Afterwards, the 30 experimental tests on T-stubs were performed, and the four failures modes identified in EC9 were observed. The experimental tests result showed that the initial stiffness and the ultimate resistance increase significantly when the distance from the pin to the web or the thickness of the flange is increased. However, decreasing the distance between the pins and the web reduced the deformation capacity. Additionally, it was also noticed that the swage-locking pins load-carrying capacity, the collar type and the pin diameter affected the structural behaviour. Finally, a new design method based on the continuous strength method (CSM) was proposed, which was validated with the experimental test results. Besides, the predictions of the EC9 method were evaluated, and It showed that the method underestimates the load capacity of extruded aluminium T-stub connections.

In 2020, Wang et al. [97] continued investigating the behaviour of extruded aluminium T-stubs connected by swage locking pins. In the previous paper [73], the investigation developed experimental tests on 30 T-stubs, which was then used to validate the FEM models of this investigation. Before starting the T-stubs analysis, the researchers developed refined and simplified FEM models of the swage-locking pins. The simulations were compared with experimental data, and they showed that the refined model reproduced with good accuracy the behaviour of the pins but with a high computational time of processing. Therefore, the simplified model was developed through the application of a four-step calibration methodology. This methodology allowed to reduce the computational time without lost sufficient accuracy. Afterwards, the FEM models, which used 8-node bricks elements with reduced integration and hourglass control (C3D8R), of the extruded T-stubs was validated by comparing the numerically derived load-carrying capacities, load-displacement curves and failure modes with those obtained from the 30 experiments [73]. Then a parametric study was performed to provide insight influence of four key parameters (the preloaded in swage-locking pins, pin diameter, fillet radius at the web-to-flange junction and pitch distance of the pins) on the structural behaviour. The simulation results showed that the T-stub's initial stiffness increased when the preloading and the fillet radio increased, but the deformation capacity was not influenced. Furthermore, load-carrying was not influenced by the preloading level. The pin diameter affected the T-stub failure mode directly due to the change in the resistance area. Finally, the authors proposed a design method that was more accurate than the EC9 formulae due to the improvement in the prediction of 13%.

In 2020, Wang et al. [98] aimed to achieve a damage-controllable and earthquake-resilient steel frame with more feasible and simple dissipative devices. Therefore, the

authors studied an innovative moment-resisting bolted connection, employing FEM, that used double replaceable T-stub fuses made of low yield point steel (LYP). Because of the higher ductility dissipation capacity and the fatigue performance [99] of the LYP steel, the connection ductility and energy dissipation capacity are enhanced. The finite element model, which was used to study the cyclic behaviour, was calibrated with available experimental data that were obtained by Iannone et al. [100] and Chou et al. [101]. Then the seismic response (hysteresis curves) of the connection, with different weakening strategies in the T-stub, were studied and optimise by considering the most relevant parameters, the weakened degree and the free deformation length, which should be simultaneously analysed. Furthermore, the authors proposed relevant suggestions for designing this kind of connections to behave as structural fuses that can dissipate more than 90% of the system energy and how to avoid the buckling in the T-stub stem.

In 2020, Yuan et al. [102] tested a total of 13 austenitic and duplex stainless-steel T-stubs connected with A4–70 and A4–80 stainless steel bolts to study the T-stub behaviour. These experimental tests were replicated by employing a 3D FEM model. After the 13 simulations reproduced with good accuracy the behaviour of the T-stubs, that were obtained in the experimental testing, a parametric study was performed to predict the behaviour of 168 configurations and to investigate the importance of key parameters such as the material grade, bolt preloading, bolt diameter and flange thickness on the structural response. The test result data and the simulation results were used to determine the suitability of the design standard EC3 and JGJ 82-2011 (Chinese standard) for steel and was determined that the stainless T-stubs had the same three failure modes of steel T-stubs, but the resistance prediction was quite conservative. Therefore, the nominal yield strength value used in the design standards to predict the T-stub behaviour was replaced by the 3.0% proof strain (due to strain hardening) and the prediction were more accurate. Additionally the method of Faella predicts more precisely the initial stiffness than the other standards method.

## 7. Discussion

Readers will now understand that the T-stub component has been widely studied since Douty and McGuire started their investigation of the T-stubs in tension. Nowadays, the study of the T-stub behaviour covers components under tension, compression, cyclic and impact loads of different types of connections. The different types of connections refer to the uses of backing plates, the type of bolt (standard, blind bolt, one side bolt) and the material of the T-stub. Additionally, many studies of the behaviour of the T-stubs at elevated temperature were developed.

The present review article divided into three manners the way T-stubs were studied in order to determine if there are new ways of approaching the problem and found (as was hoped) that the first researchers approached the problem by performing experimental tests due to the limitations of the computational technology of the time. Therefore, analytical models were proposed to study the problem. For instance, Douty and McGuire proposed some semi-empirical formulae, Agerskov proposed an analytical model of the T-stub, Zotemeijer proposed formulae that employed similar failure modes to the actual failure modes, etc. The analytical models are still being in used due to the fact their predictions can be obtained without the necessity of high computational hardware. Moreover, the codes such as Eurocode 8 and 9, AISC, etc. propose different formulae (according to the employed theory) for predict the T-stub behaviour. However, as aforementioned, the standards cannot cover all the possibilities of configuration of geometries or materials (as the last part of the review covers). Therefore, some proposed models based on EC3 (the beam theory) were presented in this review. These non-standarised models were validated against experimental data (the most reliable) and by developing FEM models. The FEM models provided the investigators with the possibility of comparing their models against numerical results of non-tested geometric configurations, which is the advantage of the numerical FEM models tools. This comparison could result in the modification of the proposed model due to the lack of accuracy for some scenarios, see Piluso [36,37] and [41]. This lack of

accuracy could be caused by different factors, such as internal forces (bending or shear), geometry simplification, etc. that were not considered. Additionally, the comparison of the models of Spyrou [24] and Heidarpour [25] against the EC3, experimental data and simulations showed that these proposed models should be improved with reliable (experimental and/or simulation) data to be more accurate. The last manner of studying the T-stub behaviour, informational models, is the least used and (considered by the authors newest) it is not common for traditional engineers who are more accustomed to analytical formulae found in the codes or employing FEM models. The informational model that was found in the literature was the one developed by Ceniceros et al. [78], and this is a hybrid approach between the FE method and metamodels based on soft computing (SC), which predicts the T-stub behaviour thanks to employing the data of FEM model and parametric study and experimental tests to teach and calibrate the informational model. As was mentioned, the informational models can developed their objective due to the neuronal networks that they employed, which is more common for computer science researchers, mathematicians, etc.

Finally, according to the information that the authors could find in their literature review, they suggest that future investigation of T-stub connected by another kind of bolts such as the Ajax Oneside ST (standard tension and shear resistance) and Oneside Hi shear (standard tension resistance and high shear resistance). This type of bolts has one hardened split/collapsible washer, which could cause that the behaviour of the connection change due to the non-use of a solid washer. Additionally, investigation about the influence of the uses of ultra-twist bolt, the BOM fastener and the Huck bolt should be performed due to the singular manners that these devices use to fasten to different elements. Regarding the material, it is suggested to investigate the T-stubs obtained by additive manufacturing of different steel materials such as high strength steel, stainless steel, aluminium, etc. and the influence of the type of additive manufacturing employed. Another important aspect of the T-stub behavior that should be further improved is its behaviour at elevated temperatures because the analytical models are not accurate enough under those conditions. Besides, it is suggested to study the T-stub under impact and thermal load, which could represent a possible scenario of storing hazardous materials such as liquid and gas fuel. Regarding the new trends in processing-analysis information, the authors recommend continuing the investigation of new informational models based on artificial neural networks in order to improve the time needed to accurately predict the T-stub behaviour. It should be noted that institutions like banks use this technology and enormous amounts of data are processed using ANN models.

**Author Contributions:** Conceptualization, G.J.B.A. and J.J.J.d.C.F.; methodology, J.J.J.d.C.F.; validation, J.H.A.M. and A.M.G.A.; investigation, J.J.J.d.C.F. and G.J.B.A.; writing—original draft preparation, G.J.B.A.; writing—review and editing, G.J.B.A.; visualization, G.J.B.A.; supervision, J.J.J.d.C.F.; project administration, J.J.J.d.C.F.; funding acquisition, J.J.J.d.C.F. All authors have read and agreed to the published version of the manuscript.

**Funding:** "This work has been developed thanks to the funding granted by the Dirección General de Investigation (DGI)" through the funds of the Annual Project Contest (CAP 2019) of the Pontifical Catholic University of Peru.

**Institutional Review Board Statement:** Not applicable.

**Informed Consent Statement:** Not applicable.

**Data Availability Statement:** This study do not report any data to support the experimental part of the review or any other part. Just the relevant information such as geometry of the T-stub, resistance, type of load, etc., were extrated from some articles of the review in order to summaries them.

**Acknowledgments:** The authors thank the Pontificia Universidad Católica del Perú for the financial support because without that this review would not be possible.

**Conflicts of Interest:** The authors declare no conflict of interest.

## Appendix A

The following link contains relevant information of the T-stubs studied by some of the authors that are presented in this review: https://docs.google.com/spreadsheets/d/1cHcrjTTuZgQxWb3hLCe5SNSwpU0Dj8-b/edit#gid=648261224 (accessed on 7 October 2021).

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
