# Peer review of "A Review of the T-Stub Components for the Analysis of Bolted Moment Joints"

_applsci, doi:10.3390/app112210731_

Round 1
Reviewer 1 Report
The paper and the topic are interesting. I have some comments before possibe acceptance.
- the state of the art and the discussion about the ultimate response of T-Stub should be improved considering what can is the effect of membrane action in the flange at very large deformation. Indeed. at very large deformation (e.g. under severe seismic action or in case of column loss), the gap opening of the flange triggers the development of catenary effects in the flange and the bolts are subjected to shear and bending more than axial forces. THerefore, I recommend to refer and discuss what is given in https://doi.org/10.1016/j.istruc.2020.08.039
- the influence of bolts can affect the behaviour of T-Stubs in mode 3. Indeed, EU preloaded bolts are available as HR and HV. The former exhibit shank necking and the latter nut stripping. Therefore, the ductility of T-Stub can change. I suggest to consider and discuss this issue on the basis of what reported in Doi: 10.1016/j.jcsr.2016.05.017
Author Response
Response to reviewer 1
Dear reviewer 1,
First of all, the authors are pretty grateful for the given comments and suggestions. These suggestions let us known the parts of the manuscripts that should be improved, and we realised that the type of the two different high resistance bolts could influence the T-stub behaviour. Therefore, the manuscript has changed, and now more crucial information has been added.
Point 1: the state of the art and the discussion about the ultimate response of T-Stub should be improved considering what can is the effect of membrane action in the flange at very large deformation. Indeed. at very large deformation (e.g. under severe seismic action or in case of column loss), the gap opening of the flange triggers the development of catenary effects in the flange and the bolts are subjected to shear and bending more than axial forces. THerefore, I recommend to refer and discuss what is given in https://doi.org/10.1016/j.istruc.2020.08.039
Response 1: The ultimate response of T-stub under large deformation was presented in section "2.5". This section added the analytical model for T-stub under large deformations of the research work "Experimental and numerical study on the T-Stub behaviour with preloaded bolts under large deformations". Additionally, in the L 716, more detailed information about the investigation developed in the aforementioned paper was summarise.
Point 2: the influence of bolts can affect the behaviour of T-Stubs in mode 3. Indeed, EU preloaded bolts are available as HR and HV. The former exhibit shank necking and the latter nut stripping. Therefore, the ductility of T-Stub can change. I suggest to consider and discuss this issue on the basis of what reported in Doi: 10.1016/j.jcsr.2016.05.017
Response 2: the aforementioned paper of Point 2 was referred to in the L 716. There was explain how the two more common high resistance bolts (HV and HR) developed different failure modes.
Finally, we want to inform you that two more research works were included. The first of the two was the recently published research work of G.J. Berrospi Aquino et al., "Resistance prediction of laminates, fillet‐welded and full penetration‐welded bolted T‐stub connections." And the last one was the recently published research work developed by J.J. Jiménez de Cisneros et al., "Meshless numerical simulation of steel connections: application to the T‐stub component". Unfortunately, the authors must delate Figures 31, Figure 19 b, Figure 20 and Figure 21 due to the fees of the copyrights, which the authors could not pay due to the lack of funds.
Best regards.
Reviewer 2 Report
The presented work contains an extensive review of the literature in the field of T-stub components analysis. The collected information was divided into three categories: analytical solutions, experimental research and numerical simulation. Based on the literature, each category is described extensively. The manuscript authors proposed analytical model for clamped T-stub based on EC3. All article point are clear and logically arranged. In presented manuscript it should by remove drawbacks as well as corrected ambiguities.
After corrected ambiguity, If the editors find it appropriate, the manuscript might be published, as no specific contraindications have been found.
Comments and ambiguities:
- The authors presented further possibilities for the development of the issue in the analytical method. Would they suggest anything for the other methods? Has only one method been focused?
- Keywords - Method of Components – it should be - Components method
- Keywords - Finite Element Simulations - incorrect definition should be either Finite element method or Numerical simulations
- Line 49 – after “mechanical spring model” please add “see, Figure 2b”
- Line 55 and Fig 3b - knee zone - please confirm that this is correct. According to the available knowledge in the field of material strength, I have not encountered this nomenclature.
- In Figures 2a and 2b the abbreviations CWS, CWT, CWC should be written in the text what these names mean
- In the manuscript the authors use the term "mods", wouldn't it be more appropriate to call it "types" or "cases"?
- The authors use the wording "Geometric properties" in the article (eg, Fig. 4, Fig. 7), "geometrical parameters" would be more appropriate.
- Figure 8 - marked in the figure “0.8r ó 0.8aÖ2” - What does it mean
- unify the names and surnames of the authors in the text. For example, in the text only the name (line 39), line 180 - model Z. Wang and line 571 - Cabaleiro M.
- There are no references to figure in the text – Fig.10; 11; 20; 21; 25.
- Editorial and punctuation errors found
- Line 106-113 - check punctuation marks
- Line 147 - is de
- Line 172 – in2016 - no spaces
- Line 183, 292 - no dot
- Line 187, 239, 240 - no spaces
- Line 250 - stiffnes matrix u2y it should be u2,y
- Line 344 – When it should be when
- Line 373 - Case 3: in this case the bolts yield first, and the prying forces are equal to cero.
- Line 563, 564 - mode ii
- Line 578 - n=15 - no spaces
- Line 812 - (dsh), (tf) – indexes
- Figure 29 - signature on the next page
- Lie 839 - (ii) deformable, it should be (iii)
Author Response
Response to reviewer 2
Dear reviewer 2,
First of all, the authors are pretty grateful for the given comments and suggestions.
Point 1: The authors presented further possibilities for the development of the issue in the analytical method. Would they suggest anything for the other methods? Has only one method been focused?
Response 1: The review is focused on presenting the analytical methods that belong to Eurocode 3. Therefore, the presented analytical models refer to EC3. However, the authors wanted to present in a brief manner the T-hanger of the AISC in order to let to know to the readers that the AISC studies something similar.
Point 2: Method of Components – it should be - Components method
Response 2: It was corrected
Point 3: Finite Element Simulations - incorrect definition should be either Finite element method or Numerical simulations
Response 3: It was changed to Numerical simulations
Point 4: Line 49 – after “mechanical spring model” please add “see, Figure 2b”
Response 4: It was added.
Point 5: Line 55 and Fig 3b - knee zone - please confirm that this is correct. According to the available knowledge in the field of material strength, I have not encountered this nomenclature.
Response 5: The nomenclature kee zone was confused with the nomenclature employed by A. Girao in her Doctoral thesis "Characterization of the ductility of bolted end plate beam-to-column steel connections" Knee range. Therefore, it was corrected and updated.
Point 6: In Figures 2a and 2b the abbreviations CWS, CWT, CWC should be written in the text what these names mean.
Response 6: The full word of the abbreviations were written between L45 to L59.
Point 7: In the manuscript the authors use the term "mods", wouldn't it be more appropriate to call it "types" or "cases"?
Response 7: This terminology is used by the EC3 and it is used by researchers such as Ana Girao, J.J. Jiménez de Cisneros, C. Faella, ETC. Therefore, the authors considered the terminology well employed.
Point 8: The authors use the wording "Geometric properties" in the article (eg, Fig. 4, Fig. 7), "geometrical parameters" would be more appropriate.
Response 8: The authors accepted the suggestion and the manuscript was updated with the proposed terminology.
Point 9: Figure 8 - marked in the figure “0.8r ó 0.8aÖ2” - What does it mean.
Response 9: The meaning of "r" and "a" were explain in L74-75
Point 10: unify the names and surnames of the authors in the text. For example, in the text only the name (line 39), line 180 - model Z. Wang and line 571 - Cabaleiro M.
Response 10: The names were unified in the format "Z. Wang"
Point 11: There are no references to figure in the text – Fig.10; 11; 20; 21; 25
Response 11: The references were added and the FIgure 20 and 21 were delete for copyright issues.
Point 12: Editorial and punctuation errors found
Response 12: The punctuation errors were corrected.
Finally, we want to inform you that two more research works were included. The first of the two was the recently published research work of G.J. Berrospi Aquino et al., "Resistance prediction of laminates, fillet‐welded and full penetration‐welded bolted T‐stub connections." And the last one was the recently published research work developed by J.J. Jiménez de Cisneros et al., "Meshless numerical simulation of steel connections: application to the T‐stub component". Unfortunately, the authors must delate Figures 31, Figure 19 b, Figure 20 and Figure 21 due to the fees of the copyrights, which the authors could not pay due to the lack of funds.
Best regards.
Reviewer 3 Report
The manuscript covers a review of T-stub components for the analysis of bolted moment joints. This topic touches upon an interesting field related to metal structures. The manuscript is written correctly but it might be a little bit too long. The title matches the manuscript content and relevant literature has been cited. To improve the manuscript authors may consider the reduction of the manuscript content to avoid its hard reading and should consider minor changes from specific comments.
Specific comments:
P 1, L 3: “connections” according to the Eurocode 3 part 8 there is a definition of connection and definition of joint. Regarding those definitions, it is suggested to revise the usage of these two words in the title and entire manuscript.
P 1, L 3: “Braced frames have a system of stiff elements called bracings, which is provided to withstand the total of the lateral forces. In the other case, the unbraced frame is adopted” – the frame with a bracing system can be classified as unbraced if the stiffness of the bracing system is not adequate. As well as the other type of classification regarding the susceptibility of the frame to second-order effects is not mentioned.
P 1, L 39: “Furthermore, this T-stub connection was first studied by Douty [3].” – The literature [3] has two authors so the citations throughout all text should be revised: Douty, R.; McGuire, W. High strength moment connections. J. Struct. Div. ASCE 1965, 91, 101–128.
P 2, L 41: “Figure 2.” The abbreviations CWT, CWS, CWC are not defined.
P 3, L 63: “Figure 3.” The fonts on the figure should be adapted to the font size of the entire manuscript.
P 3, L 65: “As a result of these investigation, the researchers got to the conclusion that the T-stub component (see Figure 4) can fail in three modes.” –figure 4 does not represent the mentioned failure modes. The mentioned failure modes were presented in Figure 5.
P 3, L 74: “The timeline organised each category” – the sentence is not clear
P 4, L 83: “Component approach (EC3) T-tub in tension” – T-stub
P 5, L 107: “beff” – eff should be in subscript. Check throughout all text also.
P 6, L 128: “2.2. Component approach (EC3) T-tub in tension with backing plates” - T-stub
P 8, L 149: “2.5. Component approach (EC9) aluminium T-tub in tension” - T-stub
P 9, L 165: “…equal to 1 if not welded in a section.” – not clear sentence
P 9, L 169: “…are the safety factor for design” – according to the Eurocodes those factors are defined as partial factors.
P 9, L 172: “…in2016.“ - space
P 21, L 398: “As a result of this, they concluded that the prying force increased the tension bolt force and identified the importance of the material strain hardening, as well.” - it is not clear who are “they” in this sentence, the authors from ref [3]?
P 24, L 479: “…experimental program tested 12 T-tub to evaluate the plastic supply.” - T-stub
P 26, L 556: “…to eliminate the flange's bending (tf=25mm), the…” – f should be in subscript
P 27, L 572: “(steel 235)” – S235?
P 28, L 600: …with great thickness),” – not clear sentence
P 29, L 642: “Figure 25.” - figure is not mentioned in the text.
P 31, L 710: “lector” - who is lector?
P 33, L 793: “T-tub” - T-stub
P 34, L 804: “…connection with bling bolt…” – blind bolt?
“…response of a cast-in-situ beam to column joint…” – not a clear sentence, the definition of the cast in situ steel structure should be provided.
P 42, L 1143: “T-tub” - T-stub
P 43, L 1198: “T-tub” - T-stub
P 43, L 1201: “T-tub” - T-stub
P 43, L 1219: “lector” - who is lector?
P 43, L 1235: “However, as it was Aforementioned…” – small letter “a”
Author Response
Response to reviewer 3
Dear reviewer 3,
First of all, the authors are pretty grateful for the given comments and suggestions.
Point 0: The manuscript covers a review of T-stub components for the analysis of bolted moment joints. This topic touches upon an interesting field related to metal structures. The manuscript is written correctly but it might be a little bit too long. The title matches the manuscript content and relevant literature has been cited. To improve the manuscript authors may consider the reduction of the manuscript content to avoid its hard reading and should consider minor changes from specific comments.
Response 0: The authors are glad to read your comment. However, the review covers an extensive number of information (authors) in a long period of time, since the first studies. Therefore, the number of pages cannot be reduced.
Point 1: P 1, L 3: “connections” according to the Eurocode 3 part 8 there is a definition of connection and definition of joint. Regarding those definitions, it is suggested to revise the usage of these two words in the title and entire manuscript.
Response 1: The definition of connection and joint were revised in the manuscript and were corrected in many parts. The correction was done according to EC3 definition and the definitions of the book "Structural Steel Semirigid Connections: theory, design and software" of C. Faella et al.
Point 2: P 1, L 3: “Braced frames have a system of stiff elements called bracings, which is provided to withstand the total of the lateral forces. In the other case, the unbraced frame is adopted” – the frame with a bracing system can be classified as unbraced if the stiffness of the bracing system is not adequate. As well as the other type of classification regarding the susceptibility of the frame to second-order effects is not mentioned.
Response 2: This part was rewritten and the second-order effects classification was added according to the book "Structural Steel Semirigid Connections: theory, design and software" of C. Faella et al.
Point 3: P 1, L 39: “Furthermore, this T-stub connection was first studied by Douty [3].” – The literature [3] has two authors so the citations throughout all text should be revised: Douty, R.; McGuire, W. High strength moment connections. J. Struct. Div. ASCE 1965, 91, 101–128
Response 3: the L 39 (now L42) was corrected and the citations too.
Point 4: P 2, L 41: “Figure 2.” The abbreviations CWT, CWS, CWC are not defined.
Response 4: The full word of the abbreviations was written between L45 to L59.
Point 5: Line 55 and Fig 3b - knee zone - please confirm that this is correct. According to the available knowledge in the field of material strength, I have not encountered this nomenclature.
Response 5: The nomenclature kee zone was confused with the nomenclature employed by A. Girao in her Doctoral thesis "Characterization of the ductility of bolted end plate beam-to-column steel connections" Knee range. Therefore, it was corrected and updated.
Point 6: In Figures 2a and 2b the abbreviations CWS, CWT, CWC should be written in the text what these names mean.
Response 6: The full word of the abbreviations were written between L45 to L59.
Point 7: P 3, L 63: “Figure 3.” The fonts on the figure should be adapted to the font size of the entire manuscript.
Response 7: The fonts in Figure 3 are Palatino Linotype.
Point 8: “As a result of these investigation, the researchers got to the conclusion that the T-stub component (see Figure 4) can fail in three modes.” –figure 4 does not represent the mentioned failure modes. The mentioned failure modes were presented in Figure 5.
Response 8: The reference was delete and now the reader will not be confused.
Point 9: P 3, L 74: “The timeline organised each category” – the sentence is not clear
Response 9: The L74 was corrected and now is the L80. and the sentence was updated to "These categories are organised in chronological order".
Point 10: P 4, L 83: “Component approach (EC3) T-tub in tension” – T-stub
Response 10: It was updated.
Point 11: P 5, L 107: “beff” – eff should be in subscript. Check throughout all text also.
Response 11: It was changed to subscript.
Point 12: P 6, L 128: “2.2. Component approach (EC3) T-tub in tension with backing plates” - T-stub
P 8, L 149: “2.5. Component approach (EC9) aluminium T-tub in tension” - T-stub
Response 12: The word T-tub was changed to T-stub.
Point 13: P 9, L 165: “…equal to 1 if not welded in a section.” – not clear sentence
Response 13: The L165 was rewritten according to EC9.
Point 14:
P 9, L 169: “…are the safety factor for design” – according to the Eurocodes those factors are defined as partial factors.
P 9, L 172: “…in2016.“ - space
Response 14:
P 9, L 169: Yes, those are partial factors, and were updated to the right terminology.
P 9, L 172: Space was added.
Point 15: P 21, L 398: “As a result of this, they concluded that the prying force increased the tension bolt force and identified the importance of the material strain hardening, as well.” - it is not clear who are “they” in this sentence, the authors from ref [3]?
Response 15: The ref [3] was added.
Point 16: P 24, L 479: “…experimental program tested 12 T-tub to evaluate the plastic supply.” - T-stub
Response 16: It as corrected to T-stub
Point 17: P 26, L 556: “…to eliminate the flange's bending (tf=25mm), the…” – f should be in subscript
Response 17: f was changed to subscript.
Point 18: P 27, L 572: “(steel 235)” – S235?
Response 18: The right name is S235.
Point 19: P 28, L 600: …with great thickness),” – not clear sentence
Response 19: It was clarified in L624-627.
Point 20: P 29, L 642: “Figure 25.” - figure is not mentioned in the text.
Response 20: Now the Figure 25 is mentioned.
Point 21: P 31, L 710: “lector” - who is lector?
Response 21: Lector was confused with the reader. Lector is a reader in Spanish
Point 22: P 33, L 793: “T-tub” - T-stub
Response 22: T-tub was changed to T-stub
Point 23: P 34, L 804: “…connection with bling bolt…” – blind bolt?
Response 23: Blind bolt it is correct. This was corrected in the manuscript
Point 24: “…response of a cast-in-situ beam to column joint…” – not a clear sentence, the definition of the cast in situ steel structure should be provided.
Response 24: A-cast-in situ beam that is made of concrete, revised L 1081-1085
Point 25:
P 42, L 1143: “T-tub” - T-stub
P 43, L 1198: “T-tub” - T-stub
P 43, L 1201: “T-tub” - T-stub
Response 25: T-tub was changed to T-stub in P42, P43 and P43
Point 26: P 43, L 1219: “lector” - who is lector?.
Response 26:The word lector is in Spanish and was used instead of reader.
Point 27: P 43, L 1235: “However, as it was Aforementioned…” – small letter “a”
Response 27: It was changed to small letter a
Finally, we want to inform you that two more research works were included. The first of the two was the recently published research work of G.J. Berrospi Aquino et al., "Resistance prediction of laminates, fillet‐welded and full penetration‐welded bolted T‐stub connections." And the last one was the recently published research work developed by J.J. Jiménez de Cisneros et al., "Meshless numerical simulation of steel connections: application to the T‐stub component". Unfortunately, the authors must delate Figures 31, Figure 19 b, Figure 20 and Figure 21 due to the fees of the copyrights, which the authors could not pay due to the lack of funds.
Best regards.
Round 2
Reviewer 1 Report
The revised manuscript is suitable for publication